# Optimal Comparator Adaptive Online Learning with Switching Cost

**Zhiyu Zhang**
Boston University
zhiyuz@bu.edu

**Ashok Cutkosky**
Boston University
ashok@cutkosky.com

**Ioannis Ch. Paschalidis**
Boston University
yannisp@bu.edu

## Abstract

Practical online learning tasks are often naturally defined on unconstrained domains, where optimal algorithms for general convex losses are characterized by the notion of *comparator adaptivity*. In this paper, we design such algorithms in the presence of switching cost – the latter penalizes the typical optimism in adaptive algorithms, leading to a delicate design trade-off. Based on a novel *dual space scaling* strategy discovered by a continuous-time analysis, we propose a simple algorithm that improves the existing comparator adaptive regret bound [ZCP22a] to the optimal rate. The obtained benefits are further extended to the expert setting, and the practicality of the proposed algorithm is demonstrated through a sequential investment task.[1]

## 1 Introduction

Online learning [CBL06, Haz16, Ora19] is a powerful framework for modeling sequential decision making tasks, such as neural network training, financial investment and robotic control. In each round, an agent picks a prediction $x_t$ in a convex domain $\mathcal{X}$, receives a convex and Lipschitz loss function $l_t$ that depends on $x_1, \ldots, x_t$, and suffers the loss $l_t(x_t)$. The goal is to ensure that in any environment, the cumulative loss of the agent is never much worse than that of any fixed decision $u \in \mathcal{X}$. That is, one aims to upper-bound the regret $\sum_{t=1}^{T} [l_t(x_t) - l_t(u)]$, for all time horizon $T \in \mathbb{N}_+$, comparator $u \in \mathcal{X}$ and loss sequence $l_1, \ldots, l_T$.

If there exists a best fixed decision $u^* = \max_x \sum_{t=1}^{T} l_t(x)$ in hindsight, then the regret with respect to any $u \in \mathcal{X}$ is dominated by the regret with respect to $u^*$. In the context of training machine learning models, $u^*$ corresponds to the model parameter that minimizes the training error. Hence, intuitively, the regret bound characterizes how fast the algorithm finds $u^*$ through training.

Most classical online learning algorithms are *minimax* in nature, only tuned to optimize the worst-case regret. For example, if the domain $\mathcal{X}$ is bounded, then the maximum distance between the best fixed decision $u^*$ and the initialization $x_1$ of the algorithm is the diameter $D$ of $\mathcal{X}$. Only considering this worst case, it suffices to use *Online Gradient Descent* (OGD) [Zin03] with learning rate $\eta_t \propto D/\sqrt{t}$. The result is a $O(D\sqrt{T})$ regret bound that holds uniformly for all comparators $u \in \mathcal{X}$. Such a minimax reasoning is prevalent, but limited in two substantial ways.

1. It requires a bounded domain. Many practical problems are naturally unconstrained, making such arguments inapplicable.

2. Practical applications are usually not the worst case, which means that the minimax bound $O(D\sqrt{T})$ is typically loose. Think about the situation where we have a prior guess of $u^*$, from either domain knowledge or pre-training. Using this prior as the initial prediction $x_1$, we should

---

[1]Future versions available at https://arxiv.org/abs/2205.06846.

36th Conference on Neural Information Processing Systems (NeurIPS 2022).

expect a provable gain over arbitrarily initializing the algorithm – specifically, had we known the *correct* distance $\|u^* - x_1\|$, we could have picked $\eta_t \propto \|u^* - x_1\| / \sqrt{t}$ in OGD, resulting in $O(\|u^* - x_1\| \sqrt{T})$ regret. Achieving this goal *without the prior knowledge of* $\|u^* - x_1\|$ is of both theoretical and practical importance.

Recent studies of *comparator adaptive* online learning[2] [LS15, OP16, CO18] aim to address these issues. The domain does not need to be bounded, and the regret bound is $\tilde{O}(d(u, x_1)\sqrt{T})$, where $d(\cdot, \cdot)$ is some suitable distance measure. Intuitively, these algorithms are both *optimistic* and *robust*: given prior information on $u^*$, we can pick $x_1$ such that $d(u^*, x_1)$, and consequently the regret bound, are both low. Meanwhile, even when our initialization $x_1$ is wrong (i.e., $d(u^*, x_1)$ is large), the regret bound is still *almost as good* (up to logarithmic factors) as that of the minimax algorithm with the best tuning in hindsight. Such properties have shown benefits in diverse applications, e.g., [OT17, JO19, vdH19].

In this paper, we extend the design of these algorithms to a classical setting with switching costs. Here the agent is penalized not only by its loss, but also by how fast it changes its predictions. Practically, switching costs are useful whenever the smooth operation of a system is favored, such as in network routing, control of electrical grid, portfolio management with transaction costs, etc. Recently they also naturally show up in online decision problems with *long term effects*, such as *nonstochastic control* [ABH+19]. With a given weight $\lambda \geq 0$ and a norm $\|\cdot\|$, our goal is to guarantee a comparator adaptive bound for the *augmented regret*

$$\sum_{t=1}^{T} [l_t(x_t) - l_t(u)] + \lambda \sum_{t=1}^{T-1} \|x_t - x_{t+1}\|.$$

While gradient descent can incorporate switching costs by simply scaling its learning rate, extending comparator adaptive algorithms is a lot harder. Just like other adaptive algorithms [DHS11, DGSS15], the key idea of comparator adaptivity is to quickly respond to the incoming information and hedge aggressively. Switching costs, on the other hand, encourage the agent to stay still. Therefore, achieving our goal requires a delicate balance between these two opposite considerations.

Similar trade-offs between adaptivity and switching costs have led to intriguing results in the past. For example, Gofer [Gof14] showed that the gradient variance adaptivity well-studied in the switching-free setting is impossible with normed switching costs, thus establishing a clear separation caused by the latter. Daniely and Mansour [DM19] showed that a common analytical technique for switching costs is incompatible to the so-called *"strong adaptivity"* (i.e., a form of adaptivity w.r.t. nonstationary comparators). Regarding comparator adaptivity, our prior work [ZCP22a] proposed the first comparator adaptive algorithm with switching costs, but the obtained regret bound does not achieve the optimality criterion of the switching-free setting. The present paper closes this gap.

## 1.1 Contribution

We develop comparator adaptive algorithms for two classical settings: ($i$) *Online Linear Optimization* (OLO) with switching cost; ($ii$) *Learning with Expert Advice* (LEA) with switching cost.

1. For one-dimensional unconstrained OLO with switching costs, assuming loss gradients $|g_t| \leq 1$ and initial prediction[3] $x_1 = 0$, we propose an algorithm that guarantees

$$\sum_{t=1}^{T} g_t(x_t - u) + \lambda \sum_{t=1}^{T-1} |x_t - x_{t+1}| \leq C\sqrt{\lambda T} + |u| \, O\left(\sqrt{\lambda T \log(C^{-1} |u|)}\right),$$

   where $C > 0$ is any hyperparameter chosen by the user (Section 2). Our bound achieves several forms of optimality with respect to $\lambda$, $|u|$ and $T$, improving the prior work [ZCP22a]. Extensions to bounded domains and general dimensional domains are presented, which include some new technical components (Appendix B).
2. Converting the above result from OLO to LEA, we demonstrate how classical conversion techniques [LS15, OP16] are *designed* to have large switching costs, and then propose a fix with a clear geometric interpretation. This leads to the first comparator adaptive algorithm for LEA with switching costs (Section 3).

---

[2]Also called *parameter-free* online learning due to historical reasons.

[3]For general $x_1$, we can replace $|u|$ in the regret bound by $|u - x_1|$.

Technically, our improvement over [ZCP22a] relies on a novel *dual space scaling* strategy. This is actually not guessed, but *systematically discovered* by a continuous-time analysis (Section 2.4), whose procedure follows another prior work of ours [ZCP22b]. In the continuous-time limit, it becomes evident what kinds of algorithmic structures from the switching-free setting are transferable to the setting with switching costs. Indeed, revealing generalizable knowledge is a key benefit of the continuous-time analysis, which was not demonstrated in [ZCP22b]. As an added bonus, both our OLO algorithm and its analysis are considerably simpler than those from [ZCP22a].

Concluding these theoretical results, our OLO algorithm is applied to a portfolio management task with transaction costs (Appendix D). Numerical results support its superiority over the existing approach [ZCP22a].

## 1.2 Related work

**Online learning basics**   Throughout this paper we will only consider linear losses. The generality of our setting is preserved, since convex losses can be reduced to linear losses through the relation $\sum_{t=1}^{T}[l_t(x_t) - l_t(u)] \leq \sum_{t=1}^{T} \langle \nabla l_t(x_t), x_t - u \rangle$ [Haz16, Ora19]. Online learning with linear losses is called *Online Linear Optimization* (OLO). As its important special case, *Learning with Expert Advice* (LEA) considers OLO on a probability simplex, but aims at a different form of regret bound due to its different geometry.

Classical minimax approaches in online learning include *Online Mirror Descent* (OMD) and *Follow the Regularized Leader* (FTRL), with *Online Gradient Descent* (OGD) being their most well-known special case. We write "gradient descent" as the minimax baseline for the ease of exposition. Moreover, both OMD and FTRL have elegant duality interpretations [Ora19, Section 6.4.1 and 7.3], involving simultaneous updates on the primal space (the domain $\mathcal{X}$) and the dual space (the space of gradients). We will exploit this duality in our analysis.

**Comparator adaptive online learning**   Also known as *parameter-freeness*, comparator adaptive online learning aims at matching the performance of the optimally-tuned gradient descent in hindsight, without knowing the correct tuning parameter. The associated regret bound can appear in different forms, depending on the specific learning setting.

1. For LEA, a comparator adaptive bound has the form $O\left(\sqrt{T \cdot \mathrm{KL}(u\|\pi)}\right)$, where $u$ and $\pi$ are distributions on the expert space representing the comparator and a user-chosen prior. Such an idea was initiated in [CFH09], and the analysis was improved and extended by a series of works [CV10, LS15, KVE15, CLW21, NBC+21, PLH22]. Notably, a comparator adaptive LEA algorithm naturally induces a bound on the $\varepsilon$-*quantile regret* – the regret with respect to the $\varepsilon$-quantile best expert. The latter is particularly meaningful when the number of experts is large. Lower bounds were considered in [NBC+21].
   We will present a nonasymptotic improvement of the $\sqrt{\mathrm{KL}}$ divergence in this paper. Frameworks that generalize root KL to $f$-*divergences* have been studied in [Alq21, NBC+21], but to our knowledge, no existing algorithm guarantees a better divergence term than root KL, even without switching costs.

2. For OLO, typical comparator adaptive bounds are $C + \|u\| O\left(\sqrt{T \log(C^{-1} \|u\|_* T)}\right)$ or $C\sqrt{T} + \|u\| O\left(\sqrt{T \log(C^{-1} \|u\|_*)}\right)$, where a prior $x_1$ can be incorporated by letting $u \leftarrow u - x_1$. These two bounds are both *Pareto-optimal* (see [ZCP22b] for a detailed explanation), as they represent different trade-offs on the *loss* (the regret at $u = x_1$) and the *asymptotic regret* (when $\|u - x_1\|$ is large). Existing works [MO14, CO18, FRS18, MK20, JC22] were mostly independent of the LEA setting, but unified views were presented in [FRS15, OP16]. Lower bounds were studied in [SM12, Ora13, ZCP22b].

**Switching cost**   Motivated by numerous applications, switching costs in online decision making have been studied from many different angles. For example, beside online learning, the online algorithm community has investigated settings like *smoothed online optimization* [CGW18, GLSW19, LQL20] and *convex body chasing* [BLLS19, Sel20], where the loss function $l_t$ is observed *before* the agent picks the prediction $x_t$. There, the switching cost is the key consideration that prevents the trivial strategy $x_t \in \arg\min_x l_t(x)$. As for online learning, an additional complication is that $x_t$ (e.g., the investment portfolio) should be selected without knowing $l_t$ (e.g., tomorrow's stock price).

Even within online learning, there are several ways to model switching costs. In cases like network routing, every switch means changing the packet route, which can be costly. Therefore, one needs a *lazy* agent whose amount of switches (or its expectation) [KV05, GVW10, AT18, CYLK20, SK21] is as low as possible – a good modeling candidate is $\mathbf{1}[x_t \neq x_{t+1}]$. Alternatively, one could take a *smooth* view [ABL+13, BCKP21, WWYZ21, ZJLY21] where the agent can perform as many switches as it wishes, as long as the cumulative distance of switching is low – in this view, switching cost can be a norm $\|x_t - x_{t+1}\|$ or its smoothed variant $\|x_t - x_{t+1}\|^2$. The present work primarily considers the $L_1$ norm switching cost motivated by the transaction cost in some financial applications. Notably, for LEA, the $L_1$ norm unifies the lazy view and the smooth view [DM19, Section 5.2].

Although switching costs have been extensively studied, existing works on the combination of adaptivity and switching cost are quite sparse. As one should carefully trade off these two opposite requirements, there have been interesting impossibility results [Gof14, DM19], highlighted in our introduction. In this regard, one should not believe that every classical adaptivity can be naturally achieved with switching costs. The present paper shows that the optimal comparator adaptivity can indeed be achieved, thus improving the suboptimal result from [ZCP22a].

**Relation to downstream problems**   More generally, incorporating switching costs amounts to considering a *history-dependent* adversary: it can pick loss functions that depend not only on the instantaneous prediction $x_t$, but also on the previous prediction $x_{t-1}$. One could further generalize this setting to *online learning with memory* [CBDS13, AHM15], where the loss depends on a fixed-length prediction history, and finally to *dynamical systems* [ABH+19, SSH20, Sim20], where the entire history matters. In fact, a common procedure in *nonstochastic control* [ABH+19] is to bound the risk in the future by a properly scaled switching cost. Achieving comparator adaptivity with switching costs may benefit these important downstream problems as well, by making algorithms *easy to combine* [Cut19, Cut20, ZCP22a].

**Continuous-time approach to online learning**   Finally, on the technical side, our methodology builds on an emerging continuous-time perspective of online learning. From a theoretical angle, Kohn and Serfaty [KS10] demonstrated a rigorous connection between *Partial Differential Equations* (PDE) and discrete-time repeated games. More recently, such a connection has led to *algorithmic benefits* in minimax LEA [Zhu14, Rok17, DK20, KKW20, HLPR20, BEZ20a, BEZ20b, GHP22], $\varepsilon$-quantile bounds [PLH22] and comparator adaptive OLO [ZCP22b]. The key idea is that online learning algorithms in the continuous-time limit can be more easily parameterized and analyzed. In this paper we will show an additional bonus: generalizing algorithmic insights is also easier in the continuous-time limit.

## 1.3   Notation

Let $f^*$ be the Fenchel conjugate of a function $f$. $\Delta(d)$ represents the $d$-dimensional probability simplex; KL and TV denote the KL divergence and the total variation distance, respectively. For two integers $a \leq b$, $[a : b]$ is the set of all integers $c$ such that $a \leq c \leq b$. log represents the natural logarithm when the base is omitted. Throughout this paper, "increasing" and "positive" are not strict (i.e., include equality as well).

For a twice differentiable function $V(t, S)$ where $t$ represents time and $S$ represents a spatial variable, let $\nabla_t V$, $\nabla_{tt} V$, $\nabla_S V$ and $\nabla_{SS} V$ be the first and second order partial derivatives. In addition, we define discrete derivatives as

$$\bar{\nabla}_t V(t, S) := V(t, S) - V(t-1, S),$$

$$\bar{\nabla}_S V(t, S) := \frac{1}{2}\left[V(t, S+1) - V(t, S-1)\right], \tag{1}$$

$$\bar{\nabla}_{SS} V(t, S) := V(t, S+1) + V(t, S-1) - 2V(t, S).$$

## 2   OLO with switching cost

This section presents our main result, a comparator adaptive OLO algorithm with switching costs. We will focus on the 1D unconstrained setting. Due to limited space, extensions to general settings are deferred to Appendix B.

## 2.1 The 1D unconstrained setting

We consider the domain $\mathcal{X} = \mathbb{R}$, a Lipschitz constant $G > 0$ for the loss gradients, and a weight $\lambda \geq 0$ for switching costs. In the $t$-th round, the agent predicts $x_t \in \mathbb{R}$, receives a loss gradient $g_t \in [-G, G]$ that depends on past predictions $x_{1:t}$, and suffers an augmented loss $g_t x_t + \lambda \left| x_t - x_{t-1} \right|$ (w.l.o.g., let $x_0 = x_1 = 0$). The performance metric is the augmented regret for all $u \in \mathbb{R}$ and $T \in \mathbb{N}_+$:

$$\mathrm{Regret}_T^\lambda(u) := \sum_{t=1}^{T} g_t(x_t - u) + \lambda \sum_{t=1}^{T-1} \left| x_t - x_{t+1} \right|. \tag{2}$$

Ignoring the dependence on $G$ for now, our goal is to show a comparator adaptive bound $\tilde{O}\left( |u| \sqrt{\lambda T} \right)$, more specifically the optimal rates $C + |u| \, O\left( \sqrt{\lambda T \log(C^{-1}\lambda |u| T)} \right)$ or $C\sqrt{\lambda T} + |u| \, O\left( \sqrt{\lambda T \log(C^{-1} |u|)} \right)$ for any hyperparameter $C > 0$. These two cases are equivalent via the standard doubling trick [SS11], as discussed in [ZCP22b].

For minimax algorithms like bounded domain gradient descent [Zin03], one can use scaled learning rates $\eta_t \propto 1/\sqrt{\lambda t}$ to ensure that both sums in (2) are $O\left( \sqrt{\lambda T} \right)$, thus obtaining a combined $O\left( \sqrt{\lambda T} \right)$ regret bound. However, such a divide-and-conquer approach does not apply to comparator adaptive algorithms, as one cannot *separately* show the desirable bound on the two sums in (2). To see this, suppose one could guarantee the second sum alone is at most $1 + |u| \, O\left( \sqrt{T \log(|u| T)} \right)$; here we only focus on the dependence on $|u|$ and $T$. Since this cumulative switching cost is an algorithmic quantity *independent of the comparator*, we can take infimum with respect to $u$ and obtain a "budget" of 1 for this sum. Following this argument, $|x_T| \leq |x_1| + \sum_{t=1}^{T-1} |x_t - x_{t+1}| = O(1)$. That is, the algorithm should only predict around the origin, which clearly leads to large regret with respect to far-away comparators, under certain loss sequences.

The challenge can be motivated in another way. As shown in [Ora19, Figure 9.1], the one-step switching cost $|x_t - x_{t+1}|$ of comparator adaptive algorithms can grow exponentially with respect to $t$, whereas such a quantity is uniformly bounded in gradient descent. In fact, the exponential growth is the key mechanism for comparator adaptive algorithms to cover an unconstrained domain fast enough (thus improving minimax algorithms). This is however problematic when switching is also penalized, as one can no longer control the switching cost by uniformly scaling $|x_t - x_{t+1}|$.

## 2.2 Switching-adjusted potential

To address these issues, one should bound the switching cost and the standard OLO regret in a *unified* framework, instead of treating them separately. Our prior work [ZCP22a] used the classical coin-betting approach from [OP16, CO18], which is included in Appendix A.1 for completeness. In the $t$-th round, the algorithm maintains a quantity $\mathrm{Wealth}_{t-1}$; by picking a *betting fraction* $\beta_t \in [0, 1]$, the prediction is set to $x_t = \beta_t \mathrm{Wealth}_{t-1}$. To further ensure low switching costs, the betting fraction $\beta_t$ is capped by a decreasing upper bound $O(1/\sqrt{t})$. Although it is analytically nontrivial, such a hard threshold is conservative, which could be the reason of our previous suboptimal result.

In contrast, the present paper follows the *potential framework* explored by a parallel line of works [MO14, FRS18, MK20, ZCP22b]. Generally, these algorithms are defined by a potential function $V(t, S)$, where $t$ represents the time index, and $S$ represents a "sufficient statistic" that summarizes the history. In each round, the algorithm computes $S_{t-1} = -\sum_{i=1}^{t-1} g_i/G$, and the prediction $x_t$ is the derivative $\nabla_S V$ evaluated at $(t, S_{t-1})$. We will specifically consider Algorithm 1, which is a variant based on the discrete derivative $\bar{\nabla}_S V$, cf. (1).

One could think of the potential framework as the dual approach of FTRL – the potential function and the regularizer are naturally Fenchel conjugates. While the FTRL analysis relies on a one-step regret bound on the *primal space* (the domain $\mathcal{X}$, cf. [Ora19, Lemma 7.1]), the potential framework constructs a similar one-step relation on the *dual space* (the space of $S_t$, cf. [ZCP22b, Lemma 3.1]). Along this interpretation, **our key idea is to incorporate switching costs by scaling on the dual space, rather than only on the primal space.** That is, given a potential function that works without switching costs, we scale the sufficient statistic sent to its second argument by a function of $\lambda$.

---

**Algorithm 1** One-dimensional unconstrained OLO with switching costs.

---

**Require:** A hyperparameter $C > 0$, the Lipschitz constant $G$, and a potential function $V(t, S)$ that implicitly depends on $\lambda$ and $G$. Initialize $S_0 = 0$.
1: **for** $t = 1, 2, \ldots$ **do**
2:     Predict $x_t = \bar{\nabla}_S V(t, S_{t-1})$, and receive the loss gradient $g_t$. Let $S_t = S_{t-1} - g_t/G$.
3: **end for**

---

To better demonstrate this idea, let us first consider a quadratic potential $V(t, S) = (1/2) \cdot CGS^2$. The potential method suggests the prediction $x_t = \nabla_S V(t, S_{t-1}) = C \sum_{i=1}^{t-1} g_i = x_{t-1} - Cg_{t-1}$, which is simply gradient descent with learning rate $C$. Scaling on the primal space means scaling $V$ directly, while scaling on the dual space means scaling the sufficient statistic $S$. It is clear that both cases are *equivalent* to scaling the effective learning rate, which is the standard way to incorporate switching costs in bounded domain gradient descent. In other words, for this gradient descent potential, the two types of scaling are essentially the same.

Now, to achieve optimal comparator adaptivity, we need a better potential where scaling on the dual space actually makes a difference. With a parameter $\alpha$ that will eventually depend on $\lambda$, we consider Algorithm 1 induced by the potential

$$V_\alpha(t, S) = C\sqrt{\alpha t} \left[ 2 \int_0^{S/\sqrt{4\alpha t}} \left( \int_0^u \exp(x^2) dx \right) du - 1 \right]. \tag{3}$$

When the Lipschitz constant $G = 1$, it has been shown [ZCP22b] that $\alpha = 1/2$ leads to comparator adaptivity without switching costs. Here we use $\alpha = 4\lambda G^{-1} + 2$, which amounts to scaling *both* the primal space and the dual space: on the primal space, we scale up the overall prediction by $\Theta(\sqrt{\lambda G^{-1} + 1})$, and on the dual space we scale down the sufficient statistic $S$ by $\Theta(1/\sqrt{\lambda G^{-1} + 1})$. The latter gives us the optimal comparator adaptive bound (i.e., Pareto-optimal rate in $|u|$ and $T$), while the former helps us obtain the optimal rate in $\lambda$. Due to incorporating $\lambda$ into the potential function $V_\alpha$, we call our approach the *switching-adjusted potential method*.

Although the dual space scaling strategy and the particular structure of $V_\alpha$ may seem mysterious at first glance, they are actually *derived* from a continuous-time analysis. To proceed, we will first present the performance guarantee in the next subsection, and then revisit the derivation of this strategy in Section 2.4.

## 2.3 Optimal comparator adaptive bound

Despite its simplicity, our approach improves the result from our prior work [ZCP22a] by a considerable margin.

**Theorem 1.** *If $\alpha = 4\lambda G^{-1} + 2$, then Algorithm 1 induced by the potential $V_\alpha$ guarantees*

$$\mathrm{Regret}_T^\lambda(u) \le \sqrt{(4\lambda G + 2G^2)T} \left[ C + |u| \left( \sqrt{4 \log \left( 1 + \frac{|u|}{C} \right)} + 2 \right) \right],$$

*for all $u \in \mathbb{R}$ and $T \in \mathbb{N}_+$.*

Theorem 1 simultaneously achieves several forms of optimality.

1. Pareto-optimal loss-regret trade-off: considering the dependence on $u$ and $T$, $\mathrm{Regret}_T^\lambda(u) = O\left(|u|\sqrt{T \log |u|}\right)$, while the *cumulative loss* satisfies $\mathrm{Regret}_T^\lambda(0) = O(\sqrt{T})$. An existing lower bound [ZCP22b, Theorem 10] shows that even without switching costs, all algorithms satisfying a $O(\sqrt{T})$ loss bound must suffer a $\Omega\left(|u|\sqrt{T \log |u|}\right)$ regret bound. In this sense, our algorithm attains a *Pareto-optimal* loss-regret trade-off, in a strictly generalized setting with switching costs.
2. On $T$ alone: $\mathrm{Regret}_T^\lambda(u) = O(\sqrt{T})$. Despite achieving comparator adaptivity, the asymptotic rate on $T$ is still the optimal one, matching the well-known minimax lower bound.
3. On $\lambda$ alone: $\mathrm{Regret}_T^\lambda(u) = O(\sqrt{\lambda})$. Our bound has the optimal dependence on the switching cost weight [GVW10, Theorem 5].

To compare Theorem 1 to [ZCP22a], we have to convert them to the same loss-regret trade-off, i.e., both guaranteeing $\mathrm{Regret}_T^\lambda(0) = O(1)$ or $\mathrm{Regret}_T^\lambda(0) = O(\sqrt{T})$. Here we take the first approach – details are presented in Appendix A.4. Let us only consider the dependence on $u$ and $T$.[4] By a doubling trick, our bound can be converted to $C + |u| O\left(\sqrt{T \log(C^{-1} |u| T)}\right)$, which improves the rate $C + |u| O\left(\sqrt{T} \log(C^{-1} |u| T)\right)$ from [ZCP22a, Theorem 1]. Specifically, our converted upper bound also attains Pareto-optimality in this regime (i.e., matching the lower bound $\Omega\left(|u| \sqrt{T \log(|u| T)}\right)$ in [Ora13]), whereas the existing approach does not.

The proof of Theorem 1 is sketched below, with the formal analysis deferred to Appendix A.3. It mostly follows a standard potential argument, which is another benefit over the existing approach – the idea of this proof is easier to interpret and generalize.

**Proof sketch of Theorem 1**  To begin with, the first step is to show a one-step bound on the growth rate of the potential. If there is no switching cost, then the *Discrete Itô formula* can serve this purpose, which applies to any convex potential $V$. It is an established result in the probability literature [Kud82, Fuj08, Kle13], and Harvey et al. [HLPR20] first applied it to minimax LEA. The version below is from our prior work, which is a small variant that removes the LEA context.

**Lemma 2.1** (Lemma 3.1 of [ZCP22b]). *If the potential function $V(t, S)$ is convex in $S$, then against any adversary, Algorithm 1 guarantees for all $t \in \mathbb{N}_+$,*

$$V(t, S_t) - V(t-1, S_{t-1}) \leq -G^{-1} g_t x_t + \bar{\nabla}_t V(t, S_{t-1}) + (1/2) \cdot \bar{\nabla}_{SS} V(t, S_{t-1}).$$

Our key observation is the following lemma, which incorporates switching costs into $V_\alpha$. Note that the structure of $V_\alpha$ is important here.

**Lemma 2.2.** *For all $\alpha > 0$, consider Algorithm 1 induced by the potential $V_\alpha$. For all $t \in \mathbb{N}_+$,*

$$|x_t - x_{t+1}| \leq \bar{\nabla}_S V_\alpha(t, S_{t-1} + 1) - \bar{\nabla}_S V_\alpha(t, S_{t-1} - 1).$$

Combining the above, if we define

$$\Delta_t := \bar{\nabla}_t V_\alpha(t, S_{t-1}) + \frac{1}{2} \bar{\nabla}_{SS} V_\alpha(t, S_{t-1}) + G^{-1} \lambda \left[ \bar{\nabla}_S V_\alpha(t, S_{t-1} + 1) - \bar{\nabla}_S V_\alpha(t, S_{t-1} - 1) \right],$$
(4)

then a telescopic sum yields a *cumulative loss bound*

$$\mathrm{Regret}_T^\lambda(0) \leq \sum_{t=1}^{T} \left( g_t x_t + \lambda |x_t - x_{t+1}| \right) \leq -G \cdot V_\alpha(T, S_T) + G \sum_{t=1}^{T} \Delta_t.$$

To proceed, we need to control the residual term $\Delta_t$, which may seem problematic due to its complicated form. Fortunately, a careful analysis shows that $\Delta_t$ *vanishes* with a proper choice of $\alpha$.

**Lemma 2.3.** *If $\alpha \geq 4\lambda G^{-1} + 2$, then for all $t$ and against any adversary, $\Delta_t \leq 0$.*

Finally, with the updated loss bound $\mathrm{Regret}_T^\lambda(0) \leq -G \cdot V_\alpha(T, S_T)$, our regret bound follows from the classical loss-regret duality [MO14, Ora19].

## 2.4 Continuous-time derivation

Now let us derive our dual space scaling strategy from a continuous-time perspective. Technically, the procedure is analogous to another prior work of ours [ZCP22b], which studies optimal potential functions for the standard OLO setting without switching costs. Before starting, we need a generalized definition of the discrete derivative, with a tunable gap increment $\delta$.

$$\bar{\nabla}_S^\delta V(t, S) = \frac{1}{2\delta} \left[ V(t, S + \delta) - V(t, S - \delta) \right].$$

Note that the choice of $\delta = 1$ recovers $\bar{\nabla}_S V(t, S)$ in Algorithm 1. The Lipschitz constant $G$ will be set to 1 for the ease of exposition.

---

[4]Comparing the dependence on $\lambda$ is more subtle, as discussed in Appendix A.1.

**Step 1: discrete-time recursive inequality**    First, let us consider the following inequality that characterizes "admissible" potentials for Algorithm 1. For all $t$ and $S$,

$$V(t-1, S) \geq \max_{g \in [-1,1]} \left\{ V(t, S-g) + g\bar{\nabla}_S^1 V(t, S) + \lambda \left| \bar{\nabla}_S^1 V(t, S) - \bar{\nabla}_S^1 V(t+1, S-g) \right| \right\}. \tag{5}$$

Finding solutions of this inequality is sufficient for constructing regret bounds. To see this, suppose the above holds for some $V$. We can then plug in $S = S_{t-1}$ and guarantee that for all $g_t \in [-1, 1]$,

$$g_t x_t + \lambda |x_t - x_{t+1}| \leq V(t-1, S_{t-1}) - V(t, S_t).$$

A telescopic sum further leads to a cumulative loss bound $\mathrm{Regret}_T^\lambda(0) \leq V(0,0) - V(T, S_T)$, and a regret bound on $\mathrm{Regret}_T^\lambda(u)$ then follows from the standard loss-regret duality [MO14].

**Step 2: $\varepsilon$-scaled recursion**    Since we ideally need *optimal potential functions* that satisfy the inequality (5) without any slack, let us turn (5) into an *equality* and try to approximately solve it. Intuitively this is a challenging task, as there is no natural way to parameterize the dependence of $V$ on the discrete time $t$. However, if we decrease the discrete time interval, solutions $V$ will be "smoother" and easier to describe. Concretely, let $\varepsilon > 0$ be a parameter that will later approach 0. On (5), we scale

1. the unit time by $\varepsilon^2$;
2. the loss gradient $g$, the switching cost weight $\lambda$ and the gap increment by $\varepsilon$.

Both scaling factors are justified in the switching-free setting [ZCP22b, Appendix A.2]. Notably, since $g$ and $\lambda$ have the same "unit", it is natural that they are scaled by the same rate. With that, we obtain a *scaled recursion*

$$\begin{aligned}
V(t - \varepsilon^2, S) \\
= \max_{g \in [-1,1]} \left\{ V(t, S - \varepsilon g) + \varepsilon g \bar{\nabla}_S^\varepsilon V(t, S) + \varepsilon \lambda \left| \bar{\nabla}_S^\varepsilon V(t, S) - \bar{\nabla}_S^\varepsilon V(t + \varepsilon^2, S - \varepsilon g) \right| \right\}. \quad (6)
\end{aligned}$$

**Step 3: continuous-time PDE**    To proceed, we take the second-order Taylor approximation on all components of (6). Calculations are simple, and we defer the details to Appendix A.5. Both the zeroth and the first order terms of $\varepsilon$ naturally vanish. Only keeping the second order terms, we have

$$\nabla_t V(t, S) + \max_{g \in [-1,1]} \left( \frac{1}{2} g^2 \nabla_{SS} V(t, S) + \lambda |g \nabla_{SS} V(t, S)| \right) = 0.$$

As typical potential functions are convex in the sufficient statistic $S$, it is reasonable to impose an additional condition $\nabla_{SS} V(t, S) \geq 0$. Then, the above becomes the 1D *backward heat equation*

$$\nabla_t V + \alpha \nabla_{SS} V = 0,$$

where $\alpha = \lambda + 1/2$. Compared to the switching-free setting [ZCP22b, Eq. 5], we obtain the same PDE, but change the *negative thermal diffusivity* $\alpha$ from $1/2$ to $1/2 + \lambda$. This concisely characterizes the effect of switching costs on the structure of the online learning problem.

**Step 4: solving the PDE**    The final step is to solve the backward heat equation. With a hyperparameter $c$, consider solutions of the form

$$V_\alpha(t, S) = t^c g\left( \frac{S}{\sqrt{4\alpha t}} \right).$$

Plug it in, the backward heat equation reduces to the Hermite *Ordinary Differential Equation* (ODE)

$$g''(z) - 2zg'(z) + 4cg(z) = 0,$$

which is *independent of $\alpha$*. This is a crucial observation, as it reveals the correct way to generalize the knowledge from the switching-free setting to the setting with switching costs. More specifically,

- In the switching-free setting, we can take a solution $g(z)$ of the Hermite ODE, plug in the argument $z = S/\sqrt{2t}$ and obtain a potential function $V_\alpha$.
- When switching costs are considered, the above derivation suggests us to take the *same* function $g(z)$ as before, and plug in a scaled argument $z = S/\sqrt{4\alpha t}$. **This is precisely dual space scaling.**

Finally, as shown in [ZCP22b], a particularly good choice of $c$ is $1/2$. Using this choice yields the switching-adjusted potential (3).

**Remark** To summarize, through this derivation we aim to demonstrate a key benefit of the continuous-time analysis: it makes the generalization of algorithmic structures easier. This was not presented in our prior work [ZCP22b], but could be useful in the broader online learning context.

Meanwhile, we do not intend to overclaim its strength – although the continuous-time analysis provides useful intuition, we ultimately care about discrete-time regret bounds. Discretizing such arguments relies on an obscure argument that has not been made concrete yet: "$V_\alpha$ derived in the continuous time also serves as a good potential in the discrete time." Indeed, verifying this property is technically nontrivial (Section 2.3), and doing so requires a slightly more conservative choice of $\alpha$ (i.e., $4\lambda + 2$) than what is suggested above.

## 2.5 Extension beyond the 1D unconstrained setting

So far we have only considered the 1D unconstrained setting. Our results can be extended to higher dimensional domains and bounded domains, which is deferred to Appendix B.

Most notably, we present an algorithm (Algorithm 5) for 1D bounded domain: if the domain has diameter $D$, then the *switching cost alone* of this algorithm is bounded by $\tilde{O}(D\sqrt{\tau})$ on *any time interval of length* $\tau$. Such a property is crucial in [ZCP22a] for the construction of a *strongly adaptive OCO with memory* algorithm. However, the proof in [ZCP22a] critically relies on hard-thresholding a betting fraction, which, as we demonstrated in Section 2.2, is suboptimal. In contrast, our new result simultaneously achieves this property and the optimal augmented regret bound.

# 3 LEA with switching cost

Our improved results can also be applied to *LEA with switching cost*, leading to the first comparator adaptive algorithm there. Conversion techniques (from OLO to LEA) without switching costs were studied in [LS15, OP16], and since then, they have become standard tools for the online learning community. Here we present a different view on this conversion problem, based on its connection to the well-known constrained domain reduction [CO18] (surveyed in Appendix B). In particular, it leads to a mechanism for incorporating switching costs, with a clear geometric interpretation.

The setting of *LEA with switching cost* is a special case of the high-dimensional OLO problem (Appendix B). Let $d$ be the number of experts, and we define the domain $\mathcal{X}$ as the $d$-dimensional probability simplex $\Delta(d)$. Loss gradients $g_t$ satisfy $\|g_t\|_\infty \leq G$, and switching costs are measured by the $L_1$ norm. The performance metric is still the augmented regret, now defined as

$$\text{Regret}_T^\lambda(u) := \sum_{t=1}^{T} \langle g_t, x_t - u \rangle + \lambda \sum_{t=1}^{T-1} \|x_t - x_{t+1}\|_1.$$

However, the main difference with OLO is the form of comparator adaptive bounds – here we aim at $\text{Regret}_T^\lambda(u) = O(\sqrt{T \cdot \text{KL}(u\|\pi)})$, where $\pi \in \Delta(d)$ is a prior chosen at the beginning. Achieving such a root KL bound relies on techniques different from the OLO setting.

Existing approaches [LS15, OP16] have the following procedure. Given a 1D OLO algorithm that predicts on $\mathbb{R}_+$, independent copies are created for each coordinate and updated using certain surrogate losses. A meta-algorithm queries the coordinate-wise predictions $\{w_{t,i}; i \in [1:d]\}$, collects them into a weight vector $w_t = [w_{t,1}, \ldots, w_{t,d}]$, and finally predicts the scaled weight $x_t = w_t/\|w_t\|_1$ on $\Delta(d)$. Despite its general success, such an approach has a discontinuity problem when switching costs are incorporated – if two consecutive weights $w_t$ and $w_{t+1}$ are both close to the origin, then simply scaling them to $\Delta(d)$ can lead to a large switching cost, even when $\|w_t - w_{t+1}\|_1$ is small. This problem is exacerbated by the typical setting[5] of $w_1 = 0$, due to the associated analysis. A graphical demonstration is provided in Figure 1 (Left).

In contrast, our solution is based on *a unified view* of the LEA-OLO reduction and the constrained domain reduction [CO18]. Starting without switching costs, we observe that the general Banach version of the latter can also convert OLO to LEA, therefore specialized techniques are not required

---

[5]When $w_t = 0$, $x_t$ can be arbitrary on $\Delta(d)$ by definition. However, as $w_t$ changes continuously w.r.t. the observed information, it could hover around 0 at some point, thus experiencing the sketched problem.

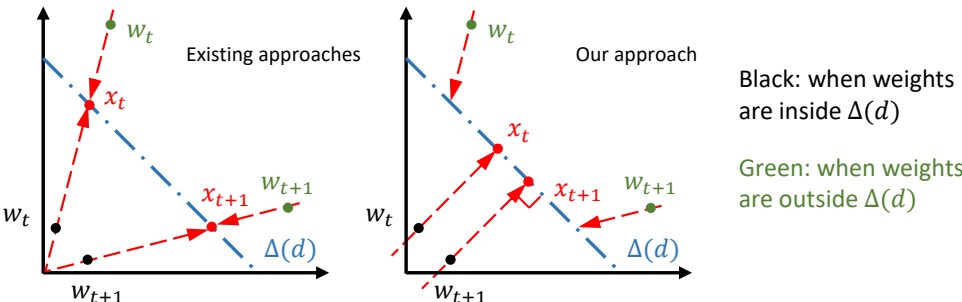

Figure 1: Switching costs in LEA-OLO reductions. Left: existing approaches. Right: ours, where the projection of $w_t$ contains two cases. $(i)$ $\|w_t\|_1 \geq 1$, shown in green; $(ii)$ $\|w_t\|_1 < 1$, shown in black.

for this task. Algorithmically, we set $x_t \in \arg\min_{x \in \Delta(d)} \|x - w_t\|_1$ as opposed to $x_t = w_t/\|w_t\|_1$. The surrogate losses for the base algorithms are also different, which we elaborate in Appendix C.3.

A major benefit of this unified view is the non-uniqueness of the $L_1$ norm projection – if $\|w_t\| < 1$, then any $x_t \in \Delta(d)$ satisfying $\{x_{t,i} \geq w_{t,i}; \forall i\}$ minimizes $\|x - w_t\|_1$ on $\Delta(d)$. This brings more flexibility to the algorithm design. Specifically, we adopt

1. the orthogonal projection $x_t = w_t + d^{-1}(1 - \|w_t\|_1)$ when $\|w_t\|_1 \leq 1$;
2. the scaling $x_t = w_t/\|w_t\|_1$ when $\|w_t\|_1 > 1$.

The orthogonal projection is better for controlling switching costs, as shown in Figure 1 (Right). Concretely, this leads to the first comparator adaptive algorithm for LEA with switching costs.

**Theorem 2.** *For LEA with switching cost, given any prior $\pi$ in the relative interior of $\Delta(d)$, Algorithm 7 from Appendix C.2 guarantees*

$$\sum_{t=1}^{T} \langle g_t, x_t - u \rangle + \lambda \sum_{t=1}^{T-1} \|x_t - x_{t+1}\|_1 = \left[ \sqrt{\mathrm{TV}(u\|\pi) \cdot \mathrm{KL}(u\|\pi)} + 1 \right] \cdot O\left( \sqrt{(\lambda G + G^2)T} \right),$$

*for all $u \in \Delta(d)$ and $T \in \mathbb{N}_+$.*

We emphasize two strengths of this bound.

1. Since it is comparator adaptive, such a bound only implicitly depends on $d$ through the divergence term $\sqrt{\mathrm{TV} \cdot \mathrm{KL}}$. In favorable cases we may have a good prior $\pi$ such that $\mathrm{TV}(u\|\pi) \cdot \mathrm{KL}(u\|\pi) = O(1)$; this will save us a $\sqrt{\log d}$ factor compared to minimax algorithms (with switching costs), such as *Follow the Lazy Leader* [KV05] and *Shrinking Dartboard* [GVW10].

2. Even without switching costs, we improve the $\sqrt{\mathrm{KL}}$ divergence term in existing comparator adaptive bounds [CFH09, LS15, OP16] to $\sqrt{\mathrm{TV} \cdot \mathrm{KL}}$. The latter is better since $(i)$ TV is always less than 1, and $(ii)$ there exist $p, q \in \Delta(d)$ such that $\mathrm{TV}(p\|q) \cdot \mathrm{KL}(p\|q) \leq 1$ but $\mathrm{KL}(p\|q) \geq \sqrt{\log d} - o(1)$ (cf. Appendix C.3). In other words, compared to $\sqrt{\mathrm{KL}}$, the $\sqrt{\mathrm{TV} \cdot \mathrm{KL}}$ bound is never worse (up to constants), and can save at least a $(\log d)^{1/4}$ factor in certain cases. Generalizations of root KL to *f-divergences* have been considered in [Alq21, NBC+21], but to our knowledge, no existing algorithm guarantees a better divergence term than root KL.

**Experiment**   We complement our theoretical results by experiments on a portfolio selection task. Due to limited space, it is presented in Appendix D.

**Conclusion**   We defer discussions on the conclusion, limitations and future works to Appendix E.

## Acknowledgments and Disclosure of Funding

We thank the anonymous reviewers for their feedback. This research was partially supported by the NSF under grants IIS-1914792, DMS-1664644, and CNS-1645681, by the ONR under grants N00014-19-1-2571 and N00014-21-1-2844, by the DOE under grants DE-AR-0001282 and DE-EE0009696, by the NIH under grants R01 GM135930 and UL54 TR004130, and by Boston University.

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
