# Appendix

**Organization**  Appendix A presents omitted details of our 1D OLO results. Section B extends such results to more general OLO settings. Section C presents details on LEA. Section D numerically tests our approach in a portfolio selection problem. Limitations and future works are discussed in Section E.

# A  Details on 1D OLO

This section presents detailed discussions and omitted proofs for our 1D unconstrained OLO algorithm.

## A.1  The suboptimal algorithm from [ZCP22a]

Let us start by summarizing the existing result from our prior work [ZCP22a, Algorithm 1], which is the first comparator adaptive algorithm for 1D unconstrained OLO with switching costs. The original version in [ZCP22a] considers a bounded domain and an extra regularization term, which are removed below for a clear comparison. $\Pi$ denotes the absolute value projection.

---

**Algorithm 2** The suboptimal algorithm from [ZCP22a]

---

**Require:**  A hyperparameter $C > 0$, the Lipschitz constant $G$, and the switching cost weight $\lambda$.
1: Define $K = G + \lambda$. Initialize internal variables as $\text{Wealth}_0 = C \cdot K$, and $\hat{\beta}_1, x_1 = 0$.
2: **for** $t = 1, 2, \ldots$ **do**
3:   Make a prediction $x_t$, observe a loss gradient $g_t$.
4:   Define an *unprojected* betting fraction as $\hat{\beta}_{t+1} = -\sum_{i=1}^t g_i/(2K^2 t)$.
5:   Define a hard threshold for the betting fraction, $\mathcal{B}_{t+1} = [-1/(K\sqrt{2t}), 1/(K\sqrt{2t})]$.
6:   Update the *projected* betting fraction as $\beta_{t+1} = \Pi_{\mathcal{B}_{t+1}}(\hat{\beta}_{t+1})$.
7:   Assign $\text{Wealth}_t$ as the solution to the following equation (uniqueness can be proved),

$$\text{Wealth}_t = (1 - g_t\beta_t)\text{Wealth}_{t-1} - \lambda|\beta_t\text{Wealth}_{t-1} - \beta_{t+1}\text{Wealth}_t|. \tag{7}$$

8:   Calculate the next prediction, $x_{t+1} = \beta_{t+1}\text{Wealth}_t$.
9: **end for**

---

Both the algorithm and its analysis are analogous to [CO18], which recasts the selection of betting fractions as an "inner" online learning problem. Nonetheless, incorporating switching costs requires extra components (Line 5-7), making the whole analysis nontrivial.

**Theorem 3** (Theorem 1 of [ZCP22a], adapted). *Algorithm 2 guarantees for all $T \in \mathbb{N}_+$ and $u \in \mathbb{R}$,*

$$\text{Regret}_T^\lambda(u) \leq (G + \lambda)\left[C + |u|\sqrt{2T}\left(\frac{3}{2} + \log\frac{\sqrt{2}\,|u|\,T^{5/2}}{C}\right)\right].$$

For now, let us only consider the dependence on $u$ and $T$. Compared to typical results on comparator adaptivity, the above bound has two limitations. First, the bound does not achieve the optimal loss-regret trade-off [ZCP22a] – the constraint $\text{Regret}_T^\lambda(0) \leq O(1)$ on the *cumulative loss* of the algorithm is too harsh, such that the leading regret term ($\text{Regret}_T^\lambda(u)$ with large $|u|$) suffers a logarithmic penalty on $T$ (relative to the usual $O(\sqrt{T})$ minimax rate). Second, even if we only consider this particular loss-regret trade-off, i.e., $\text{Regret}_T^\lambda(0) \leq O(1)$, the logarithmic terms are not optimal (being outside the square root). In other words, the bound is not Pareto-optimal. The present paper simultaneously improves these two suboptimalities.

On a separate note, let us consider its dependence on $G$ and $\lambda$, which is more subtle.[6] In its vanilla form, the above bound has the leading term $\tilde{O}(\max\{G, \lambda\}\,|u|\,\sqrt{T})$, but we can run a meta-algorithm

---

[6]We thank the NeurIPS reviewer KR3f for insightful comments.

[ZCP22a, Algorithm 3] on the top to improve it to $\tilde{O}\left(|u|\sqrt{\max\{G,\lambda\}\sum_{t=1}^{T}|g_t|}\right)$. The pseudo-code is presented as Algorithm 3. Its main idea is to adaptively slow down the update of the base algorithm, depending on the observed gradients.

---

**Algorithm 3** Meta-algorithm [ZCP22a, Algorithm 3], adapted

---

**Require:** The Lipschitz constant $G$ and the switching cost weight $\lambda$.
1: Initialize a base algorithm $\mathcal{A}$ as a copy of Algorithm 2.
2: Initialize $i = 1$ and an accumulator $Z_i = 0$. Query the first output of $\mathcal{A}$ and assign it to $w_i$.
3: **for** $t = 1, 2, \dots$ **do**
4:     Predict $x_t \leftarrow w_i$, observe $g_t$, let $Z_i \leftarrow Z_i + g_t$.
5:     **if** $\|Z_i\| > \max\{\lambda, G\}$ **then**
6:         Send $Z_i$ to $\mathcal{A}$ as the $i$-th loss. Let $i \leftarrow i + 1$.
7:         Set $Z_i = 0$. Query the $i$-th output of $\mathcal{A}$ and assign it to $w_i$.
8:     **end if**
9: **end for**

---

**Theorem 4** (Theorem 6 of [ZCP22a], adapted). *Algorithm 3 guarantees for all $T \in \mathbb{N}_+$ and $u \in \mathbb{R}$,*

$$\mathrm{Regret}_T^\lambda(u) \leq (G + \lambda)C + |u|\,\tilde{O}\left(\max\{G,\lambda\} + \sqrt{\max\{G,\lambda\}\sum_{t=1}^{T}|g_t|}\right).$$

When $\lambda$ is large, the $O(\sqrt{\lambda})$ rate on the leading term is optimal. Moreover, in the presence of switching costs, the gradient adaptivity term $\sqrt{\sum_{t=1}^{T}|g_t|}$ is a strong one, since second-order gradient adaptivity $\sqrt{\sum_{t=1}^{T}|g_t|^2}$ is not achievable [Gof14]. Note that we can also run this meta-algorithm on top of our improved base algorithm (Algorithm 1), such that the latter achieves gradient adaptivity as well. Due to this reason, when comparing the results of the present work to [ZCP22a], we mostly leave the dependence on $\lambda$ and the gradient adaptivity out of the comparison.

## A.2 A few basic lemmas

Before proving our main result (Theorem 1), we present a few basic lemmas on Algorithm 1 and the potential function $V_\alpha$ (3), which will be useful later on. The first lemma shows the monotonicity of the discrete derivative strategy, which is quite intuitive.

**Lemma A.1.** *If the potential $V(t, S)$ is even and convex in $S$, then $\bar{\nabla}_S V(t, S)$ is odd and monotonically increasing in $S$.*

*Proof of Lemma A.1.* $\bar{\nabla}_S V(t, S)$ is odd due to the simple relation

$$\bar{\nabla}_S V(t, -S) = \frac{1}{2}\left[V(t, -S + 1) - V(t, -S - 1)\right]$$
$$= \frac{1}{2}\left[V(t, S - 1) - V(t, S + 1)\right] = -\bar{\nabla}_S V(t, S).$$

As for the monotonicity, it is equivalent to showing for all $\delta \geq 0$,

$$V(t, S + 1 + \delta) - V(S - 1 + \delta) \geq V(t, S + 1) - V(S - 1).$$

This follows from the convexity of $V(t, \cdot)$, as

$$V(t, S + 1) \leq \frac{2}{2 + \delta}V(t, S + 1 + \delta) + \frac{\delta}{2 + \delta}V(t, S - 1),$$

$$V(t, S - 1 + \delta) \leq \frac{\delta}{2 + \delta}V(t, S + 1 + \delta) + \frac{2}{2 + \delta}V(t, S - 1). \qquad \square$$

Now, for the potential function $V_\alpha$, we compute its continuous partial derivatives. The proof is straightforward calculation, therefore omitted.

**Lemma A.2.** *For any $\alpha > 0$, $V_\alpha$ defined in (3) is even and convex. Moreover,*

$$\nabla_S V_\alpha(t, S) = C \int_0^{S/\sqrt{4\alpha t}} \exp\left(x^2\right) dx, \qquad \nabla_{SSS} V_\alpha(t, S) = \frac{CS}{4(\alpha t)^{3/2}} \exp\left(\frac{S^2}{4\alpha t}\right),$$

$$\nabla_{SS} V_\alpha(t, S) = \frac{C}{2\sqrt{\alpha t}} \exp\left(\frac{S^2}{4\alpha t}\right), \qquad \nabla_t V_\alpha(t, S) = -\frac{C\sqrt{\alpha}}{2\sqrt{t}} \exp\left(\frac{S^2}{4\alpha t}\right).$$

Based on the above, the discrete derivative $\bar{\nabla}_S V_\alpha$ has the following properties.

**Lemma A.3.** *For all $\alpha > 0$, $t \geq 0$ and $S \geq 0$,*

1. $\bar{\nabla}_S V_\alpha(t, S)$ *as a function of $t$ is decreasing and convex;*

2. $\bar{\nabla}_S V_\alpha(t, S)$ *as a function of $S$ is convex.*

*Proof of Lemma A.3.* Considering the first part of the lemma,

$$\nabla_t \left[\bar{\nabla}_S V_\alpha(t, S)\right] = \frac{1}{2} \left[\nabla_t V_\alpha(t, S+1) - \nabla_t V_\alpha(t, S-1)\right]$$
$$= -\frac{C\sqrt{\alpha}}{4\sqrt{t}} \exp\left(\frac{S^2+1}{4\alpha t}\right) \sinh\left(\frac{S}{2\alpha t}\right),$$

which, when $S \geq 0$, is negative and increasing in $t$. Therefore, $\bar{\nabla}_S V_\alpha(t, S)$ as a function of $t$ is decreasing and convex. Similarly,

$$\nabla_S \left[\bar{\nabla}_S V_\alpha(t, S)\right] = \frac{1}{2} \left[\nabla_S V_\alpha(t, S+1) - \nabla_S V_\alpha(t, S-1)\right] = \frac{C}{2} \int_{(S-1)/\sqrt{4\alpha t}}^{(S+1)/\sqrt{4\alpha t}} \exp\left(x^2\right) dx,$$

which is increasing in $S$. Therefore, $\bar{\nabla}_S V_\alpha(t, S)$ as a function of $S$ is convex. $\qquad\square$

### A.3 Proof of Theorem 1

In this subsection, we prove Theorem 1, the regret bound of our 1D OLO algorithm with switching costs. As sketched in Section 2.3, our proof relies on two important lemmas, Lemma 2.2 and 2.3. We prove them first.

**Lemma 2.2.** *For all $\alpha > 0$, consider Algorithm 1 induced by the potential $V_\alpha$. For all $t \in \mathbb{N}_+$,*

$$|x_t - x_{t+1}| \leq \bar{\nabla}_S V_\alpha(t, S_{t-1} + 1) - \bar{\nabla}_S V_\alpha(t, S_{t-1} - 1).$$

*Proof of Lemma 2.2.* First, since $\bar{\nabla}_S V_\alpha(t, S)$ is monotonic in $S$ due to Lemma A.1, we have

$$|x_t - x_{t+1}| = \left|\bar{\nabla}_S V_\alpha(t, S_{t-1}) - \bar{\nabla}_S V_\alpha(t+1, S_t)\right|$$
$$\leq \max_{c=\pm 1} \left|\bar{\nabla}_S V_\alpha(t, S_{t-1}) - \bar{\nabla}_S V_\alpha(t+1, S_{t-1} + c)\right|.$$

For clarity, from the RHS we define

$$f(t, S) := \max_{c=\pm 1} \left|\bar{\nabla}_S V_\alpha(t, S) - \bar{\nabla}_S V_\alpha(t+1, S+c)\right|.$$

It is even in $S$, as

$$f(t, -S) = \max_{c=\pm 1} \left|\bar{\nabla}_S V_\alpha(t, -S) - \bar{\nabla}_S V_\alpha(t+1, -S+c)\right|$$
$$= \max_{c=\pm 1} \left|-\bar{\nabla}_S V_\alpha(t, S) + \bar{\nabla}_S V_\alpha(t+1, S-c)\right| \qquad \text{(Lemma A.1)}$$
$$= \max_{c=\pm 1} \left|\bar{\nabla}_S V_\alpha(t, S) - \bar{\nabla}_S V_\alpha(t+1, S-c)\right| = f(t, S).$$

Therefore, we can restrict the rest of the proof to $S \geq 0$, and the remaining task is to upper bound $f(t, S)$ for all $0 \leq S \leq t - 1$.

From Lemma A.1 and A.3,

$$\bar{\nabla}_S V_\alpha(t+1, S-1) \leq \bar{\nabla}_S V_\alpha(t+1, S) \leq \bar{\nabla}_S V_\alpha(t, S),$$

$$\bar{\nabla}_S V_\alpha(t+1, S-1) \le \bar{\nabla}_S V_\alpha(t+1, S+1).$$

Therefore, if $\bar{\nabla}_S V_\alpha(t+1, S-1) \le \bar{\nabla}_S V_\alpha(t, S) \le \bar{\nabla}_S V_\alpha(t+1, S+1)$, then

$$f(t, S) = \max\left\{ \left| \bar{\nabla}_S V_\alpha(t, S) - \bar{\nabla}_S V_\alpha(t+1, S-1) \right|, \left| \bar{\nabla}_S V_\alpha(t, S) - \bar{\nabla}_S V_\alpha(t+1, S+1) \right| \right\}$$
$$\le \bar{\nabla}_S V_\alpha(t+1, S+1) - \bar{\nabla}_S V_\alpha(t+1, S-1).$$

On the other hand, if $\bar{\nabla}_S V_\alpha(t+1, S+1) \le \bar{\nabla}_S V_\alpha(t, S)$, then

$$f(t, S) = \bar{\nabla}_S V_\alpha(t, S) - \bar{\nabla}_S V_\alpha(t+1, S-1).$$

Combining the above,

$$f(t, S)$$
$$\le \max\left\{ \bar{\nabla}_S V_\alpha(t, S) - \bar{\nabla}_S V_\alpha(t+1, S-1), \bar{\nabla}_S V_\alpha(t+1, S+1) - \bar{\nabla}_S V_\alpha(t+1, S-1) \right\}.$$

Our goal next is to upper bound $f(t, S)$ by $\bar{\nabla}_S V_\alpha(t, S+1) - \bar{\nabla}_S V_\alpha(t, S-1)$, which can be divided into two cases.

**Case 1** We aim to show that

$$\bar{\nabla}_S V_\alpha(t, S) - \bar{\nabla}_S V_\alpha(t+1, S-1) \le \bar{\nabla}_S V_\alpha(t, S+1) - \bar{\nabla}_S V_\alpha(t, S-1),$$

which is equivalent to

$$\bar{\nabla}_S V_\alpha(t, S-1) - \bar{\nabla}_S V_\alpha(t+1, S-1) \le \bar{\nabla}_S V_\alpha(t, S+1) - \bar{\nabla}_S V_\alpha(t, S). \tag{8}$$

Note that this trivially holds when $0 \le S < 1$: due to Lemma A.3, the RHS is always positive; however, the LHS is negative due to $\bar{\nabla}_S V_\alpha(t, S-1)$ being increasing in $t$ (Lemma A.1 and A.3 Part 1). Therefore, we only need to show (8) for all $S \ge 1$.

To this end, with $S \ge 1$, we apply the convexity of $\bar{\nabla}_S V_\alpha$ from Lemma A.3:

$$\bar{\nabla}_S V_\alpha(t, S+1) - \bar{\nabla}_S V_\alpha(t, S) \ge \nabla_S \left[ \bar{\nabla}_S V_\alpha(t, S) \right],$$

$$\bar{\nabla}_S V_\alpha(t, S-1) - \bar{\nabla}_S V_\alpha(t+1, S-1) \le -\nabla_t \left[ \bar{\nabla}_S V_\alpha(t, S-1) \right].$$

Consequently, it suffices to show that

$$-\nabla_t \left[ \bar{\nabla}_S V_\alpha(t, S-1) \right] \le \nabla_S \left[ \bar{\nabla}_S V_\alpha(t, S) \right].$$

Now it is time to invoke the specific form of $V_\alpha$. We may reuse $\nabla_S \left[ \bar{\nabla}_S V_\alpha(t, S) \right]$ and $\nabla_t \left[ \bar{\nabla}_S V_\alpha(t, S) \right]$ calculated from the proof of Lemma A.3.

$$\nabla_S \left[ \bar{\nabla}_S V_\alpha(t, S) \right] = \frac{C}{2} \int_{(S-1)/\sqrt{4\alpha t}}^{(S+1)/\sqrt{4\alpha t}} \exp\left( x^2 \right) dx \ge \frac{C}{2\sqrt{\alpha t}} \exp\left( \frac{S^2}{4\alpha t} \right),$$

and for all $1 \le S \le t - 1$,

$$-\nabla_t \left[ \bar{\nabla}_S V_\alpha(t, S-1) \right] = \frac{C\sqrt{\alpha}}{4\sqrt{t}} \exp\left( \frac{(S-1)^2 + 1}{4\alpha t} \right) \sinh\left( \frac{S-1}{2\alpha t} \right)$$
$$= \frac{C\sqrt{\alpha}}{8\sqrt{t}} \exp\left( \frac{S^2}{4\alpha t} \right) \left[ 1 - \exp\left( \frac{-S+1}{\alpha t} \right) \right]$$
$$\le \frac{C\sqrt{\alpha}}{8\sqrt{t}} \exp\left( \frac{S^2}{4\alpha t} \right) \left[ 1 - \exp\left( -\frac{1}{\alpha} \right) \right] \qquad (S - 1 \le t)$$
$$\le \frac{C}{8\sqrt{\alpha t}} \exp\left( \frac{S^2}{4\alpha t} \right). \qquad (\exp(x) \ge x + 1)$$

Therefore, $-\nabla_t \left[ \bar{\nabla}_S V_\alpha(t, S-1) \right] \le \nabla_S \left[ \bar{\nabla}_S V_\alpha(t, S) \right]$, which proves (8) and concludes Case 1.

**Case 2** We aim to show that

$$\bar{\nabla}_S V_\alpha(t+1, S+1) - \bar{\nabla}_S V_\alpha(t+1, S-1) \leq \bar{\nabla}_S V_\alpha(t, S+1) - \bar{\nabla}_S V_\alpha(t, S-1).$$

This is straightforward, as

$$
\begin{aligned}
&\nabla_t \left[ \bar{\nabla}_S V_\alpha(t, S+1) - \bar{\nabla}_S V_\alpha(t, S-1) \right] \\
&= \frac{1}{2} \left[ \nabla_t V_\alpha(t, S+2) + \nabla_t V_\alpha(t, S-2) - 2 \nabla_t V_\alpha(t, S) \right] \\
&= -\frac{C\sqrt{\alpha}}{4\sqrt{t}} \left[ \exp\left( \frac{(S+2)^2}{4\alpha t} \right) + \exp\left( \frac{(S-2)^2}{4\alpha t} \right) - 2\exp\left( \frac{S^2}{4\alpha t} \right) \right] \\
&\leq 0. \qquad\qquad\qquad\qquad\qquad\qquad\qquad\qquad\qquad\qquad \text{(convexity)}
\end{aligned}
$$

Combining the two cases, we can upper bound $f(t, S)$ by $\bar{\nabla}_S V_\alpha(t, S+1) - \bar{\nabla}_S V_\alpha(t, S-1)$, which completes the proof. $\qquad\square$

Next, we present the proof of Lemma 2.3, which bounds the residual term $\Delta_t$ defined in (4).

**Lemma 2.3.** *If $\alpha \geq 4\lambda G^{-1} + 2$, then for all $t$ and against any adversary, $\Delta_t \leq 0$.*

*Proof of Lemma 2.3.* We restate the definition of $\Delta_t$ for easier reference.

$$\Delta_t = \bar{\nabla}_t V_\alpha(t, S_{t-1}) + \frac{1}{2} \bar{\nabla}_{SS} V_\alpha(t, S_{t-1}) + G^{-1}\lambda \left[ \bar{\nabla}_S V_\alpha(t, S_{t-1}+1) - \bar{\nabla}_S V_\alpha(t, S_{t-1}-1) \right].$$

Let us define a function $g(t, S)$ as

$$
\begin{aligned}
g(t, S) := &\frac{1}{2} V_\alpha(t, S+1) + \frac{1}{2} V_\alpha(t, S-1) - V_\alpha(t-1, S) \\
&+ \frac{\lambda}{2G} \left[ V_\alpha(t, S+2) + V_\alpha(t, S-2) - 2V_\alpha(t, S) \right],
\end{aligned}
$$

then from the definition of discrete derivatives, $\Delta_t = g(t, S_{t-1})$. Also note that $g(t, S)$ is even in $S$, so we can only focus on positive values of $S$. The rest of the proof will show $g(t, S) \leq 0$ for all $t \in \mathbb{N}_+$ and $S \geq 0$.

Let us start from the special case, $t = 1$. $S$ can only take the value 0, therefore $g(1, S) = g(1, 0)$. We now present a general result that upper bounds $g(t, 0)$ for all $t \in \mathbb{N}_+$:

$$
\begin{aligned}
&g(t, 0) \\
&= V_\alpha(t, 1) - V_\alpha(t-1, 0) + G^{-1}\lambda V_\alpha(t, 2) - G^{-1}\lambda V_\alpha(t, 0) \\
&= C\sqrt{\alpha t} \left[ 2 \int_0^{1/\sqrt{4\alpha t}} \left( \int_0^u \exp(x^2) dx \right) du + \frac{2\lambda}{G} \int_0^{1/\sqrt{\alpha t}} \left( \int_0^u \exp(x^2) dx \right) du + \sqrt{\frac{t-1}{t}} - 1 \right] \\
&\leq C\sqrt{\alpha t} \left[ 2 \cdot \frac{1}{2} \cdot \frac{1}{\sqrt{4\alpha t}} \int_0^{1/\sqrt{4\alpha t}} \exp(x^2) dx + \frac{2\lambda}{G} \cdot \frac{1}{2} \cdot \frac{1}{\sqrt{\alpha t}} \int_0^{1/\sqrt{\alpha t}} \exp(x^2) dx + \sqrt{\frac{t-1}{t}} - 1 \right] \\
&\qquad\qquad\qquad\qquad\qquad\qquad\qquad\qquad\qquad\qquad\quad \text{(erfi}(x) \text{ is increasing and convex on } \mathbb{R}_+) \\
&\leq C\sqrt{\alpha t} \left[ \frac{1}{4\alpha t} \exp\left( \frac{1}{4\alpha t} \right) + \frac{\lambda}{G\alpha t} \exp\left( \frac{1}{\alpha t} \right) + \sqrt{\frac{t-1}{t}} - 1 \right].
\end{aligned}
$$

Since $\sqrt{1+x} \leq 1 + x/2$ for all $x \geq -1$, we have $\sqrt{(t-1)/t} - 1 \leq -1/(2t)$. Therefore, if $\alpha \geq 4\lambda G^{-1} + 2$, then

$$
\begin{aligned}
g(t, 0) &\leq C\sqrt{\alpha t} \left[ \frac{\lambda G^{-1} + (1/4)}{\alpha t} \exp\left( \frac{1}{\alpha t} \right) - \frac{1}{2t} \right] \\
&\leq \frac{C\sqrt{\alpha}}{\sqrt{t}} \left[ \frac{\lambda G^{-1} + (1/4)}{\alpha} \exp\left( \frac{1}{2} \right) - \frac{1}{2} \right] \leq 0. \quad (9)
\end{aligned}
$$

As its special case, we have $g(1, 0) \leq 0$, which concludes the proof of the special case ($t = 1$).

Next, we prove $g(t, S) \leq 0$ for general $t$, i.e., $t \geq 2$. Our overall strategy is to show that for all $0 \leq S \leq t-1$, $g(t, S) \leq g(t, 0)$, and then using the argument above we have $g(t, 0) \leq 0$. Concretely, let us calculate the first and second order derivatives of $g(t, S)$, using Lemma A.2.

$$\nabla_S g(t, S)$$
$$= \frac{C}{2} \left[ \int_0^{(S+1)/\sqrt{4\alpha t}} \exp(x^2) dx + \int_0^{(S-1)/\sqrt{4\alpha t}} \exp(x^2) dx - 2 \int_0^{S/\sqrt{4\alpha(t-1)}} \exp(x^2) dx \right]$$
$$+ \frac{\lambda C}{2G} \left[ \int_0^{(S+2)/\sqrt{4\alpha t}} \exp(x^2) dx + \int_0^{(S-2)/\sqrt{4\alpha t}} \exp(x^2) dx - 2 \int_0^{S/\sqrt{4\alpha t}} \exp(x^2) dx \right],$$

$$\nabla_{SS} g(t, S)$$
$$= \frac{C}{4\sqrt{\alpha t}} \left[ \frac{\lambda}{G} \exp\left( \frac{(S+2)^2}{4\alpha t} \right) + \exp\left( \frac{(S+1)^2}{4\alpha t} \right) - \frac{2\lambda}{G} \exp\left( \frac{S^2}{4\alpha t} \right) \right.$$
$$\left. + \exp\left( \frac{(S-1)^2}{4\alpha t} \right) + \frac{\lambda}{G} \exp\left( \frac{(S-2)^2}{4\alpha t} \right) \right] - \frac{C}{2\sqrt{\alpha(t-1)}} \exp\left( \frac{S^2}{4\alpha(t-1)} \right)$$
$$= \frac{C}{2\sqrt{\alpha t}} \exp\left( \frac{S^2}{4\alpha t} \right) \left[ \frac{\lambda}{G} \exp\left( \frac{1}{\alpha t} \right) \cosh\left( \frac{S}{\alpha t} \right) + \exp\left( \frac{1}{4\alpha t} \right) \cosh\left( \frac{S}{2\alpha t} \right) \right.$$
$$\left. - \frac{\lambda}{G} - \sqrt{\frac{t}{t-1}} \exp\left( \frac{S^2}{4\alpha t(t-1)} \right) \right]. \tag{10}$$

Notice that $\nabla_S g(t, 0) = 0$. To proceed, we aim to prove $\nabla_{SS} g(t, S) \leq 0$ for all $S \geq 0$, which then shows $g(t, S) \leq g(t, 0)$. To this end, we will show the sum inside the bracket in (10) is negative. Denote it as $h(t, S)$, and more specifically,

$$h(t, S)$$
$$:= \frac{\lambda}{G} \exp\left( \frac{1}{\alpha t} \right) \cosh\left( \frac{S}{\alpha t} \right) + \exp\left( \frac{1}{4\alpha t} \right) \cosh\left( \frac{S}{2\alpha t} \right) - \frac{\lambda}{G} - \sqrt{\frac{t}{t-1}} \exp\left( \frac{S^2}{4\alpha t(t-1)} \right).$$

The rest of the proof is divided into two steps: we first prove $(i)$ $h(t, 0) \leq 0$; and then prove $(ii)$ $\nabla_S h(t, S) \leq 0$ for all $S \geq 0$.

**Step 1: prove $h(t, 0) \leq 0$.** From the definition of $h(t, S)$,

$$h(t, 0) = \frac{\lambda}{G} \exp\left( \frac{1}{\alpha t} \right) + \exp\left( \frac{1}{4\alpha t} \right) - \frac{\lambda}{G} - \sqrt{\frac{t}{t-1}}.$$

Letting $x = 1/t$, then to prove $h(t, 0) \leq 0$ for all $t \geq 2$, it suffices to prove

$$\psi(x) := \frac{\lambda}{G} \exp\left( \frac{x}{\alpha} \right) + \exp\left( \frac{x}{4\alpha} \right) - \frac{\lambda}{G} - \sqrt{\frac{1}{1-x}} \leq 0,$$

on the range $x \in (0, 1/2]$. $\psi(0) = 0$, and

$$\nabla_x \psi(x) = \frac{\lambda}{\alpha G} \exp\left( \frac{x}{\alpha} \right) + \frac{1}{4\alpha} \exp\left( \frac{x}{4\alpha} \right) - \frac{1}{2}(1-x)^{-3/2}$$
$$\leq \frac{4\lambda G^{-1} + 1}{4\alpha} \exp\left( \frac{1}{2\alpha} \right) - \frac{1}{2},$$

which is negative when $\alpha \geq 4\lambda G^{-1} + 2$. Therefore, $h(t, 0) \leq 0$ for all $t \geq 2$.

**Step 2: prove $\nabla_S h(t, S) \leq 0$.** Taking the derivative of $h(t, S)$,

$$\nabla_S h(t, S) = \frac{\lambda}{\alpha t G} \exp\left(\frac{1}{\alpha t}\right) \sinh\left(\frac{S}{\alpha t}\right) + \frac{1}{2\alpha t} \exp\left(\frac{1}{4\alpha t}\right) \sinh\left(\frac{S}{2\alpha t}\right)$$

$$- \sqrt{\frac{t}{t-1}} \cdot \frac{S}{2\alpha t(t-1)} \exp\left(\frac{S^2}{4\alpha t(t-1)}\right)$$

$$\leq \left(\frac{\lambda}{\alpha t G} + \frac{1}{2\alpha t}\right) \exp\left(\frac{1}{\alpha t}\right) \sinh\left(\frac{S}{\alpha t}\right) - \frac{S}{2\alpha t^2} \sqrt{\frac{t}{t-1}}.$$

Note that for all $x$, $\exp(-x) \geq 1 - x$, therefore for all $0 \leq x < 1$, $\exp(x/2) \leq \sqrt{1/(1-x)}$. Assigning $x$ to $1/t$ which is less than 1, we have for all $\alpha \geq 2$,

$$\exp\left(\frac{1}{\alpha t}\right) \leq \exp\left(\frac{1}{2t}\right) \leq \sqrt{\frac{t}{t-1}}.$$

Moreover, for all $0 \leq x \leq 1$, $\sinh(x) \leq 2x$. Therefore,

$$\nabla_S h(t, S) \leq \sqrt{\frac{t}{t-1}} \left[\frac{\lambda G^{-1} + (1/2)}{\alpha t} \sinh\left(\frac{S}{\alpha t}\right) - \frac{S}{2\alpha t^2}\right] \leq \frac{S}{\alpha^2 t^2} \sqrt{\frac{t}{t-1}} \left[2\lambda G^{-1} + 1 - \frac{\alpha}{2}\right].$$

When $\alpha \geq 4\lambda G^{-1} + 2$, $\nabla_S h(t, S) \leq 0$ for all $t \geq 2$ and $S \geq 0$.

Concluding the above two steps, we have shown $h(t, S) \leq 0$. Plugging it back into (10), we have $\nabla_{SS} g(t, S) \leq 0$, which shows that for all $t \geq 2$ and $S \geq 0$, $g(t, S) \leq g(t, 0)$. Finally, $g(t, 0) \leq 0$ following (9). $\qquad\square$

Now, given the two important lemmas above (Lemma 2.2 and 2.3), our Theorem 1 follows from a standard loss-regret duality. Details are presented below.

**Theorem 1.** *If $\alpha = 4\lambda G^{-1} + 2$, then Algorithm 1 induced by the potential $V_\alpha$ guarantees*

$$\text{Regret}_T^\lambda(u) \leq \sqrt{(4\lambda G + 2G^2)T} \left[C + |u| \left(\sqrt{4\log\left(1 + \frac{|u|}{C}\right)} + 2\right)\right],$$

*for all $u \in \mathbb{R}$ and $T \in \mathbb{N}_+$.*

*Proof of Theorem 1.* Combining Lemma 2.1, 2.2 and 2.3, we have

$$\sum_{t=1}^{T} \left(g_t x_t + \lambda |x_t - x_{t+1}|\right) \leq -G \cdot V_\alpha(T, S_T).$$

Consider $V_\alpha(T, S_T)$ as a function of $S_T$; let us write $V_{\alpha,T}^*(\cdot)$ as its Fenchel conjugate. Then, the augmented regret can be bounded as

$$\text{Regret}_T^\lambda(u) = \sum_{t=1}^{T} g_t(x_t - u) + \lambda \sum_{t=1}^{T-1} |x_t - x_{t+1}|$$

$$\leq G \cdot u S_T + \sum_{t=1}^{T} \left(g_t x_t + \lambda |x_t - x_{t+1}|\right)$$

$$\leq G\left[u S_T - V_\alpha(T, S_T)\right] \leq G \cdot V_{\alpha,T}^*(u).$$

The last step is to bound $V_{\alpha,T}^*(u)$, which also follows from a standard proof strategy.

$$V_{\alpha,T}^*(u) = \sup_{S \in \mathbb{R}} uS - V_\alpha(T, S).$$

It is clear that the supremum is uniquely achieved; let $S^*$ be the maximizing argument. Then,

$$u = \nabla_S V_\alpha(T, S^*) = C \int_0^{S^*/\sqrt{4\alpha T}} \exp\left(x^2\right) dx.$$

If we define $\mathrm{erfi}(z) = \int_0^z \exp(x^2)dx$ (note that it a scaled version of the conventional *imaginary error function*), then $S^* = \sqrt{4\alpha T} \cdot \mathrm{erfi}^{-1}\left(uC^{-1}\right)$.

$$V_{\alpha,T}^*(u) = uS^* - V_\alpha(T, S^*) \leq uS^* - V_\alpha(T, 0) = C\sqrt{\alpha T} + |u|\sqrt{4\alpha T} \cdot \mathrm{erfi}^{-1}\left(uC^{-1}\right).$$

Finally, as shown in [ZCP22b, Theorem 4], $\mathrm{erfi}^{-1}(x) \leq 1 + \sqrt{\log(1+x)}$. Combining the above completes the proof. $\qquad\square$

### A.4  Conversion of loss-regret trade-offs

In this subsection we discuss the conversion of loss-regret trade-offs in unconstrained OLO. Our Theorem 1 guarantees a loss bound $\mathrm{Regret}_T^\lambda(0) = O(\sqrt{T})$ and an accompanying regret bound $\mathrm{Regret}_T^\lambda(u) = O\left(|u|\sqrt{T\log|u|}\right)$. By a doubling trick (effectively, a meta-algorithm), we can turn such guarantees into $\mathrm{Regret}_T^\lambda(0) = O(1)$ and $\mathrm{Regret}_T^\lambda(u) = O\left(|u|\sqrt{T\log(|u|T)}\right)$. These can be directly compared to [ZCP22a]. Concretely, we present the classical doubling trick as Algorithm 4.

---

**Algorithm 4** Conversion of loss-regret trade-offs.

---

**Require:** A hyperparameter $C > 0$, and a base unconstrained OLO algorithm $\mathcal{A}$. Here we define $\mathcal{A}$ as the algorithm considered in Theorem 1, with $\alpha = 8\lambda G^{-1} + 2$.
1: **for** $m = 0, 1, 2, \ldots$ **do**
2:     Initialize a copy of $\mathcal{A}$ as $\mathcal{A}_m$, whose hyperparameter is set to $C \cdot 2^{-m}$.
3:     Run $\mathcal{A}_m$ for $2^m$ rounds: $t = 2^m, 2^m + 1, \ldots, 2^{m+1} - 1$.
4: **end for**

---

**Theorem 5.** *Let $\alpha = 8\lambda G^{-1} + 2$. With any hyperparameter $C > 0$, Algorithm 4 guarantees for all $u \in \mathbb{R}$ and $T \in \mathbb{N}_+$,*

$$\mathrm{Regret}_T^\lambda(u) \leq \frac{\sqrt{2\alpha}G}{\sqrt{2} - 1}\left[C + |u|\sqrt{T}\left(\sqrt{8\log\left(1 + \frac{|u|T}{C}\right)} + 2\sqrt{2}\right)\right].$$

*Proof of Theorem 5.* Algorithm 4 divides the time axis into intervals of doubling lengths. On the $m$-th interval, following Theorem 1, Algorithm 4 guarantees

$$\sum_{t=2^m}^{2^{m+1}-1}\left[g_t(x_t - u) + \lambda|x_t - x_{t+1}|\right]$$

$$\leq \sum_{t=2^m}^{2^{m+1}-1} g_t(x_t - u) + 2\lambda \sum_{t=2^m}^{2^{m+1}-2}|x_t - x_{t+1}| \qquad (x_{2^{m+1}} = x_{2^m} = 0; \text{ Triangle inequality})$$

$$\leq \sqrt{\alpha}G\left[\frac{C}{\sqrt{2^m}} + |u|\sqrt{2^m}\left(\sqrt{4\log\left(1 + \frac{|u|\cdot 2^m}{C}\right)} + 2\right)\right].$$

Now consider any time horizon $T$.

$$\mathrm{Regret}_T^\lambda(u) \leq \sum_{m=0}^{\lceil\log_2 T\rceil}\sum_{t=2^m}^{2^{m+1}-1}\left[g_t(x_t - u) + \lambda|x_t - x_{t+1}|\right]$$

$$\leq \sqrt{\alpha}G\sum_{m=0}^{\lceil\log_2 T\rceil}\left[\frac{C}{\sqrt{2^m}} + |u|\sqrt{2^m}\left(\sqrt{4\log\left(1 + \frac{|u|\cdot 2^m}{C}\right)} + 2\right)\right]$$

$$\leq \sqrt{\alpha}G\left[C\sum_{m=0}^{\lceil\log_2 T\rceil}\left(\frac{1}{\sqrt{2}}\right)^m + |u|\left(\sqrt{4\log\left(1 + \frac{|u|T}{C}\right)} + 2\right)\sum_{m=0}^{\lceil\log_2 T\rceil}\sqrt{2^m}\right]$$

$$\leq \frac{\sqrt{2\alpha}G}{\sqrt{2} - 1}\left[C + |u|\sqrt{T}\left(\sqrt{8\log\left(1 + \frac{|u|T}{C}\right)} + 2\sqrt{2}\right)\right]. \qquad\square$$

Let us compare it to Theorem 3, i.e., [ZCP22a, Theorem 1], which guarantees

$$\text{Regret}_T^\lambda(u) \le (G + \lambda) \left[ C + |u| \sqrt{2T} \left( \frac{3}{2} + \log \frac{\sqrt{2}\,|u|\,T^{5/2}}{C} \right) \right].$$

If we only care about the dependence on $|u|$ and $T$, then with the same loss bound $\text{Regret}_T^\lambda(0) = O(1)$, our algorithm improves the regret bound $\text{Regret}_T^\lambda(u)$ from $O(|u|\sqrt{T}\log(|u|\,T))$ to $O(|u|\sqrt{T\log(|u|\,T)})$. The latter matches a lower bound [Ora13], therefore achieves Pareto-optimality in this regime.

### A.5 Details on the continuous-time derivation

In Step 3 of Section 2.4, we need to perform second-order Taylor approximations on the scaled recursion (6). Details are included here for completeness.

$$V(t - \varepsilon^2, S) = V(t, S) - \varepsilon^2 \nabla_t V(t, S) + o(\varepsilon^2),$$

$$V(t, S - \varepsilon g) = V(t, S) - \varepsilon g \nabla_S V(t, S) + \frac{1}{2}\varepsilon^2 g^2 \nabla_{SS} V(t, S) + o(\varepsilon^2),$$

$$\bar{\nabla}_S^\varepsilon V(t, S) = \frac{1}{2\varepsilon}\left[ V(t, S + \varepsilon) - V(t, S - \varepsilon) \right] = \nabla_S V(t, S) + o(\varepsilon),$$

$$\bar{\nabla}_S^\varepsilon V(t + \varepsilon^2, S - \varepsilon g) = \frac{1}{2\varepsilon}\left[ V(t + \varepsilon^2, S - \varepsilon g + \varepsilon) - V(t + \varepsilon^2, S - \varepsilon g - \varepsilon) \right],$$

where

$$V(t + \varepsilon^2, S - \varepsilon g + \varepsilon) = V(t, S) + \varepsilon^2 \nabla_t V(t, S) + (1 - g)\varepsilon \nabla_S V(t, S)$$
$$+ \frac{1}{2}(1 - g)^2 \varepsilon^2 \nabla_{SS} V(t, S) + o(\varepsilon^2),$$

$$V(t + \varepsilon^2, S - \varepsilon g - \varepsilon) = V(t, S) + \varepsilon^2 \nabla_t V(t, S) + (-1 - g)\varepsilon \nabla_S V(t, S)$$
$$+ \frac{1}{2}(1 + g)^2 \varepsilon^2 \nabla_{SS} V(t, S) + o(\varepsilon^2).$$

## B  Extension to general OLO settings

This section presents extensions of our 1D unconstrained OLO algorithm to more general settings.

### B.1 Constrained domain

First, consider a constrained domain $\mathcal{X} \subset \mathbb{R}$. We can use a well-known black-box reduction [CO18, Cut20] on top of our 1D unconstrained algorithm (Algorithm 1), such that the *exact* bound in Theorem 1 carries over (w.r.t. any constrained comparator $u \in \mathcal{X}$). Concretely, the pseudo-code is shown as Algorithm 5, where $\Pi$ denotes the absolute value projection in 1D.

Similar strategies apply to higher-dimensional problems, but here we emphasize the 1D special case due to an additional feature: if the domain $\mathcal{X}$ has a finite diameter $D$, then the switching cost alone of the combined algorithm has a $\tilde{O}(D\sqrt{\tau})$ bound on any time interval of length $\tau$. This could be useful when switching costs have high priority [SK21, WWYZ21] and should be independently bounded. Moreover, it allows the combination of comparator adaptive algorithms [ZCP22a] in settings with long term prediction effects (e.g., switching cost or memory).

**Theorem 6.** *Let $x^*$ be an arbitrary point in $\mathcal{X}$. For all $C > 0$, Algorithm 5 guarantees*

$$\text{Regret}_T^\lambda(u) \le \sqrt{(4\lambda G + 2G^2)T} \left[ C + |u - x^*| \left( \sqrt{4\log\left(1 + \frac{|u - x^*|}{C}\right)} + 2 \right) \right],$$

*for all $u \in \mathcal{X}$ and $T \in \mathbb{N}_+$. Moreover, if $\mathcal{X}$ has a finite diameter $D$, then on any time interval $[T_1 : T_2] \subset \mathbb{N}_+$, the same algorithm guarantees*

$$\sum_{t=T_1}^{T_2 - 1} |x_t - x_{t+1}| \le 22\sqrt{T_2 - T_1} \left[ 2D + C + 2D\sqrt{\log(1 + DC^{-1})} \right].$$

**Algorithm 5** 1D constrained OLO with switching costs.

---

**Require:** A hyperparameter $C > 0$, a closed and convex domain $\mathcal{X} \subset \mathbb{R}$, and an unconstrained algorithm $\mathcal{A}$ (Algorithm 1 induced by $V_{4\lambda G^{-1}+2}$ and the hyperparameter $C$). Let $x^*$ be an arbitrary point in $\mathcal{X}$.
1: **for** $t = 1, 2, \ldots$ **do**
2:      Query $\mathcal{A}$ for its prediction $\tilde{x}_t$.
3:      Predict $x_t = \Pi_{\mathcal{X}}(\tilde{x}_t + x^*)$ and receive a loss gradient $g_t$.
4:      Define a surrogate loss gradient $\tilde{g}_t$ as

$$\tilde{g}_t = \begin{cases} g_t, & \text{if } g_t(\tilde{x}_t + x^*) \geq g_t x_t, \\ 0, & \text{otherwise,} \end{cases}$$

     and send $\tilde{g}_t$ to $\mathcal{A}$.
5: **end for**

---

Before presenting the proof, let us discuss its technical significance. Typically, the constrained-to-unconstrained reduction is used as a black box. However, the second part of Theorem 6 relies on a *non-black-box* use of this approach – we characterize how this reduction implicitly controls the unconstrained base algorithm, resulting in the "concentration" of its sufficient statistic $S_t$ (i.e., $S_t = O(\sqrt{t})$), as if losses are stochastic. Such an observation could be of independent interest.

*Proof of Theorem 6.* The first part of the theorem directly follows from [Cut20, Theorem 2] and the contraction property of 1D projections. As for the second part, we divide the proof into five steps.

**Step 1: the "concentration" of $S_t$**    Without loss of generality, assume $S_{t-1} \geq 0$. Considering the prediction $\tilde{x}_t = \bar{\nabla}_S V_\alpha(t, S_{t-1})$ of the unconstrained base algorithm, there are two cases.

- **Case 1: $\tilde{x}_t \leq D$.**   Due to convexity,

$$\tilde{x}_t = \bar{\nabla}_S V_\alpha(t, S_{t-1}) = C\sqrt{\alpha t} \int_{(S_{t-1}-1)/\sqrt{4\alpha t}}^{(S_{t-1}+1)/\sqrt{4\alpha t}} \left( \int_0^u \exp(x^2) dx \right) du \geq C \int_0^{S_{t-1}/\sqrt{4\alpha t}} \exp(x^2) dx.$$

Similar to the proof of Theorem 1, if we define $\mathrm{erfi}(z) = \int_0^z \exp(x^2) dx$, then $S_{t-1} \leq \sqrt{4\alpha t} \cdot \mathrm{erfi}^{-1}(DC^{-1})$. As for the next round, $|S_t| \leq S_{t-1} + |g_t|/G \leq \sqrt{4\alpha t} \cdot \mathrm{erfi}^{-1}(DC^{-1}) + 1$.

- **Case 2: $\tilde{x}_t > D$.**   In this case, since $x^* \in \mathcal{X}$, we have $\tilde{x}_t + x^*$ larger than the maximum element of $\mathcal{X}$, leading to $\tilde{x}_t + x^* > x_t$. Due to the definition of the surrogate loss, $\tilde{g}_t \geq 0$. Therefore, $|S_t| \leq \max\{|S_{t-1}|, |\tilde{g}_t/G|\} \leq \max\{|S_{t-1}|, 1\}$.

Combining the two cases and their analogous arguments for $S_{t-1} \leq 0$, we can see that for all $t$, $|S_t| \leq \max\left\{\sqrt{4\alpha t} \cdot \mathrm{erfi}^{-1}(DC^{-1}) + 1, |S_{t-1}|, 1\right\}$. By induction, we obtain for all $t$,

$$|S_t| \leq \sqrt{4\alpha t} \cdot \mathrm{erfi}^{-1}(DC^{-1}) + 1.$$

**Step 2: bounding the switching cost using $S_t$**    Still, assume $S_{t-1} \geq 0$ without loss of generality. From Lemma 2.2,

$$
\begin{aligned}
&|x_t - x_{t+1}| \\
&\leq \bar{\nabla}_S V_\alpha(t, S_{t-1} + 1) - \bar{\nabla}_S V_\alpha(t, S_{t-1} - 1) \\
&= C\sqrt{\alpha t} \left[ \int_{S_{t-1}/\sqrt{4\alpha t}}^{(S_{t-1}+2)/\sqrt{4\alpha t}} \left( \int_0^u \exp(x^2) dx \right) du - \int_{(S_{t-1}-2)/\sqrt{4\alpha t}}^{S_{t-1}/\sqrt{4\alpha t}} \left( \int_0^u \exp(x^2) dx \right) du \right] \\
&\leq C\sqrt{\alpha t} \left[ \frac{2}{\sqrt{4\alpha t}} \int_0^{(S_{t-1}+2)/\sqrt{4\alpha t}} \exp(x^2) dx - \frac{2}{\sqrt{4\alpha t}} \int_0^{(S_{t-1}-2)/\sqrt{4\alpha t}} \exp(x^2) dx \right] \\
&= C \int_{(S_{t-1}-2)/\sqrt{4\alpha t}}^{(S_{t-1}+2)/\sqrt{4\alpha t}} \exp(x^2) dx \leq \frac{2C}{\sqrt{\alpha t}} \exp\left( \frac{(S_{t-1}+2)^2}{4\alpha t} \right).
\end{aligned}
$$

**Step 3: plug in the concentration of $S_t$** Next, we use the upper bound on $S_{t-1}$ to show that $|x_t - x_{t+1}| = O(Ct^{-1/2} \exp[(\text{erfi}^{-1}(DC^{-1}))^2])$. To this end, we discuss two cases regarding how the "concentration" bound (i.e., $O(\sqrt{t})$) compares to the trivial bound (i.e., $S_t \leq t$).

- **Case 1:** $\sqrt{4\alpha t} \cdot \text{erfi}^{-1}(DC^{-1}) \geq t$. In this case, note that $S_{t-1} + 1 \leq t$ and $\alpha \geq 2$,

$$|x_t - x_{t+1}| \leq \frac{2C}{\sqrt{\alpha t}} \exp\left(\frac{(S_{t-1} + 2)^2}{4\alpha t}\right) = \frac{2C}{\sqrt{\alpha t}} \exp\left(\frac{S_{t-1}^2}{4\alpha t}\right) \exp\left(\frac{4S_{t-1} + 4}{4\alpha t}\right)$$

$$\leq \frac{2\sqrt{e}C}{\sqrt{\alpha t}} \exp\left(\frac{S_{t-1}^2}{4\alpha t}\right).$$

Since $\sqrt{4\alpha t} \cdot \text{erfi}^{-1}(DC^{-1}) \geq t$, we have $t \leq 4\alpha(\text{erfi}^{-1}(DC^{-1}))^2$. Therefore,

$$\exp\left(\frac{S_{t-1}^2}{4\alpha t}\right) \leq \exp\left(\frac{t}{4\alpha}\right) \leq \exp\left[\left(\text{erfi}^{-1}(DC^{-1})\right)^2\right],$$

$$|x_t - x_{t+1}| \leq \frac{2\sqrt{e}C}{\sqrt{\alpha t}} \exp\left[\left(\text{erfi}^{-1}(DC^{-1})\right)^2\right].$$

- **Case 2:** $\sqrt{4\alpha t} \cdot \text{erfi}^{-1}(DC^{-1}) < t$. Plugging the $O(\sqrt{t})$ bound for $S_{t-1}$ into $|x_t - x_{t+1}|$,

$$|x_t - x_{t+1}| \leq \frac{2C}{\sqrt{\alpha t}} \exp\left(\frac{(\sqrt{4\alpha(t-1)} \cdot \text{erfi}^{-1}(DC^{-1}) + 3)^2}{4\alpha t}\right)$$

$$\leq \frac{2C}{\sqrt{\alpha t}} \exp\left[\left(\text{erfi}^{-1}(DC^{-1})\right)^2\right] \exp\left(\frac{6t + 9}{4\alpha t}\right)$$

$$\leq \frac{2e^2 C}{\sqrt{\alpha t}} \exp\left[\left(\text{erfi}^{-1}(DC^{-1})\right)^2\right].$$

Combining the above, we have

$$|x_t - x_{t+1}| \leq \frac{2e^2 C}{\sqrt{\alpha t}} \exp\left[\left(\text{erfi}^{-1}(DC^{-1})\right)^2\right].$$

**Step 4: upper-bound $\exp[(\text{erfi}^{-1}(x))^2]$.** Let us consider a related function $\int_0^x \text{erfi}(u)du$. Using integration by parts,

$$\int_0^x \text{erfi}(u)du = u \cdot \text{erfi}(u)\Big|_{u=0}^x - \int_0^x u \exp(u^2)du$$

$$= x \cdot \text{erfi}(x) - \frac{1}{2}\exp(x^2) + \frac{1}{2}.$$

Plugging in $x = \text{erfi}^{-1}(DC^{-1})$, we have

$$\exp\left[\left(\text{erfi}^{-1}(DC^{-1})\right)^2\right] = 2DC^{-1} \cdot \text{erfi}^{-1}(DC^{-1}) + 1 - 2\int_0^{\text{erfi}^{-1}(DC^{-1})} \text{erfi}(u)du$$

$$\leq 2DC^{-1} \cdot \text{erfi}^{-1}(DC^{-1}) + 1.$$

Then, as we did in Theorem 1, we plug in $\text{erfi}^{-1}(x) \leq 1 + \sqrt{\log(1 + x)}$ and obtain

$$|x_t - x_{t+1}| \leq \frac{11}{\sqrt{t}}\left\{2D\left[1 + \sqrt{\log(1 + DC^{-1})}\right] + C\right\}.$$

**Step 5: final bits.** Note that

$$\sum_{t=T_1}^{T_2-1} \frac{1}{\sqrt{t}} \leq \int_{T_1-1}^{T_2-1} \frac{1}{\sqrt{x}}dx \leq 2\sqrt{T_2 - 1} - 2\sqrt{T_1 - 1} \leq 2\sqrt{T_2 - T_1}.$$

Combining it with our bound on $|x_t - x_{t+1}|$ completes the proof. $\qquad\square$

## B.2 Higher dimensional domain

Now we present generalizations of our 1D algorithm to higher dimensional domains. We will primarily consider switching costs measured by the $L_1$ norm. This serves as a nice bridge towards our LEA approach and financial applications. Extensions to other norms, e.g., $L_2$ norm, is sketched at the end.

Concretely, let the domain $\mathcal{X} = \mathbb{R}^d$, $\|g_t\|_\infty \le G$, and the switching costs are measured by the $L_1$ norm. We run Algorithm 1 on each coordinate separately [SM12], and scale the hyperparameter $C$ by $1/d$. The pseudo-code is presented as Algorithm 6.

---

**Algorithm 6** $d$-dimensional OLO with $L_1$ norm switching costs.

---

**Require:** A hyperparameter $C > 0$ and Algorithm 1.
1: For each dimension $i \in [1:d]$, initialize a copy of Algorithm 1 as $\mathcal{A}_i$. It uses the hyperparameter $C/d$ and our potential $V_\alpha$, with $\alpha = 4\lambda G^{-1} + 2$.
2: **for** $t = 1, 2, \ldots$ **do**
3:     For all $i$, query $\mathcal{A}_i$ and assign its prediction to $x_{t,i}$. Define a vector as $x_t = [x_{t,1}, \ldots, x_{t,d}] \in \mathbb{R}^d$.
4:     Predict $x_t$ and receive a loss gradient $g_t = [g_{t,1}, \ldots, g_{t,d}]$.
5:     For all $i$, send $g_{t,i}$ to $\mathcal{A}_i$ as a new surrogate loss gradient.
6: **end for**

---

**Theorem 7.** *For all $C > 0$, Algorithm 6 guarantees ($\alpha = 4\lambda G^{-1} + 2$)*

$$\sum_{t=1}^{T} \langle g_t, x_t - u \rangle + \lambda \sum_{t=1}^{T-1} \|x_t - x_{t+1}\|_1 \le G\sqrt{\alpha T} \left[ C + \|u\|_1 \left( \sqrt{4 \log \left( 1 + \frac{\|u\|_\infty d}{C} \right)} + 2 \right) \right],$$

*for all $u \in \mathbb{R}^d$ and $T \in \mathbb{N}_+$.*

*Proof of Theorem 7.* We simply combine the regret on each coordinate:

$$\begin{aligned}
&\sum_{t=1}^{T} \langle g_t, x_t - u \rangle + \lambda \sum_{t=1}^{T-1} \|x_t - x_{t+1}\|_1 \\
&= \sum_{i=1}^{d} \left[ \sum_{t=1}^{T} g_{t,i}(x_{t,i} - u_i) + \lambda \sum_{t=1}^{T-1} |x_{t,i} - x_{t+1,i}| \right] \\
&\le \sqrt{(4\lambda G + 2G^2)T} \sum_{i=1}^{d} \left[ \frac{C}{d} + |u_i| \left( \sqrt{4 \log \left( 1 + \frac{|u_i| d}{C} \right)} + 2 \right) \right] \\
&\le \sqrt{(4\lambda G + 2G^2)T} \left[ C + \|u\|_1 \left( \sqrt{4 \log \left( 1 + \frac{\|u\|_\infty d}{C} \right)} + 2 \right) \right]. \qquad \square
\end{aligned}$$

As for $L_2$ norm switching costs, we can follow the polar decomposition technique from [ZCP22a, Section 2.2], which uses our 1D unconstrained OLO algorithm as the base algorithm. The only required modification is that the base algorithm should account for an extra regularization term. Concretely, instead of bounding the augmented regret (2), we should bound

$$\sum_{t=1}^{T} g_t(x_t - u) + \lambda \sum_{t=1}^{T-1} |x_t - x_{t+1}| + \frac{\gamma}{\sqrt{t}} \sum_{t=1}^{T} |x_t|,$$

for any given weight $\gamma$.

To this end, we can consider the Online Convex Optimization problem with switching costs, where the loss functions are defined by $l_t(x) = g_t x + \gamma t^{-1/2} |x|$. Such a loss function is Lipschitz, therefore we can use the OCO-OLO reduction, and run our Algorithm 1 on its linearized surrogate. Details are omitted.

## C  Details on LEA

In this section we present techniques that extend our 1D OLO algorithm to LEA with switching costs. We show that with a streamlined analysis, the general Banach version of the constrained domain reduction [CO18] can already convert 1D OLO algorithms to LEA, thus appears to be more general than the specialized techniques [LS15, OP16]. Our approach is presented as Algorithm 7.

---

**Algorithm 7** Converting OLO to LEA via the constrained domain reduction.

---

**Require:** A prior $\pi = [\pi_1, \ldots, \pi_d]$ in the relative interior of $\Delta(d)$, and Algorithm 5.
1: For each dimension $i \in [1 : d]$, initialize a copy of Algorithm 5 as $\mathcal{A}_i$. We set $\tilde{\lambda} = 4\lambda$ and $\tilde{G} = 2G$ as the switching cost weight and the Lipschitz constant considered by $A_i$. Moreover, $A_i$ uses the domain $\mathcal{X} = \mathbb{R}_+$, the offset $x^* = \pi_i$, the hyperparameter $\pi_i$, and our potential $V_\alpha$, where $\alpha = 4\tilde{\lambda}\tilde{G}^{-1} + 2$.
2: **for** $t = 1, 2, \ldots$ **do**
3:     For all $i$, query $\mathcal{A}_i$ and assign its prediction to $w_{t,i}$. Define the weight vector as $w_t = [w_{t,1}, \ldots, w_{t,d}] \in \mathbb{R}_+^d$.
4:     Compute the LEA prediction $x_t = [x_{t,1}, \ldots, x_{t,d}]$ from

$$x_{t,i} = \frac{w_{t,i} + \frac{1}{d}\max\{0, 1 - \|w_t\|_1\}}{\max\{\|w_t\|_1, 1\}}.$$

5:     Predict $x_t$ and receive a loss vector $g_t \in [-G, G]^d$.
6:     For all $i$, compute

$$z_{t,i} = \begin{cases} g_{t,i} - \|g_t\|_\infty, & \text{if } \|w_t\|_1 < 1, \\ g_{t,i}, & \text{if } \|w_t\|_1 = 1, \\ g_{t,i} + \|g_t\|_\infty, & \text{if } \|w_t\|_1 > 1, \end{cases}$$

    and return $z_{t,i}$ to $\mathcal{A}_i$ as a new surrogate loss.
7: **end for**

---

### C.1  An auxiliary lemma

Before presenting the performance guarantee of Algorithm 7, we first prove an auxiliary lemma.

**Lemma C.1.** *For all $x \geq 0$,*

$$|x - 1|\log(1 + |x - 1|) \leq 2(1 - x + x\log x).$$

*Proof of Lemma C.1.* Define LHS $-$ RHS as $h(x)$. Clearly, $h(1) = 0$. When $x > 1$,

$$h'(x) = 1 - \log x - x^{-1}.$$

It equals $0$ when $x = 1$, and $h''(x) = (1 - x)/x^2$ which is negative for all $x > 1$. Therefore, $h(x) \leq 0$ for all $x \geq 1$.

As for the case of $x < 1$,

$$h'(x) = -\log(2 - x) - \frac{1 - x}{2 - x} - 2\log x,$$

$$h''(x) = -\frac{x^2 - x + 2}{(x - 2)^2 x} < 0,$$

therefore $h(x) \leq 0$ for all $0 \leq x \leq 1$. $\qquad\square$

### C.2  Analysis of Algorithm 7

Next, we present our result on LEA with switching cost.

**Theorem 2.** *For LEA with switching cost, given any prior $\pi$ in the relative interior of $\Delta(d)$, Algorithm 7 from Appendix C.2 guarantees*

$$\sum_{t=1}^{T} \langle g_t, x_t - u \rangle + \lambda \sum_{t=1}^{T-1} \|x_t - x_{t+1}\|_1 = \left[ \sqrt{\mathrm{TV}(u\|\pi) \cdot \mathrm{KL}(u\|\pi)} + 1 \right] \cdot O\left( \sqrt{(\lambda G + G^2)T} \right),$$

*for all $u \in \Delta(d)$ and $T \in \mathbb{N}_+$.*

*Proof of Theorem 2.* We divide the proof into three steps.

**Step 1** The first step is to show that ($i$) for all $u \in \Delta(d)$, $\langle g_t, x_t - u \rangle \leq \langle z_t, w_t - u \rangle$; and ($ii$) $\|x_t - x_{t+1}\|_1 \leq O(\|w_t - w_{t+1}\|_1)$. In this way, we can translate the LEA problem to a $d$-dimensional OLO problem with the loss vector $z_t$, despite not achieving the root KL bound yet.

To prove the goal ($i$), we consider two cases, $\|w_t\|_1 < 1$ and $\|w_t\|_1 > 1$ (the case of $\|w_t\|_1 = 1$ trivially holds). If $\|w_t\|_1 < 1$, we have $x_t = w_t + d^{-1}(1 - \|w_t\|_1)$ and $z_t = g_t - \|g_t\|_\infty$.

$$\langle g_t, x_t - u \rangle = \langle g_t, w_t - u \rangle + (1 - \|w_t\|_1) \left( \sum_i g_{t,i}/d \right),$$

$$\langle z_t, w_t - u \rangle = \langle g_t, w_t - u \rangle + (1 - \|w_t\|_1) \|g_t\|_\infty,$$

therefore $\langle g_t, x_t - u \rangle \leq \langle z_t, w_t - u \rangle$. As for the case of $\|w_t\|_1 > 1$, similarly, $x_t = w_t/\|w_t\|_1$, $z_t = g_t + \|g_t\|_\infty$, $\langle g_t, x_t - u \rangle = \langle g_t, w_t/\|w_t\|_1 - u \rangle$, and $\langle z_t, w_t - u \rangle = \langle g_t, w_t - u \rangle + \|g_t\|_\infty (\|w_t\|_1 - 1)$.

$$\langle g_t, x_t - u \rangle - \langle z_t, w_t - u \rangle = - (\langle g_t, x_t \rangle + \|g_t\|_\infty)(\|w_t\|_1 - 1) \leq 0.$$

Now consider the goal ($ii$). To avoid cluttered notations, define $a_t = w_t + d^{-1} \max\{0, 1 - \|w_t\|_1\}$ and $A_t = \max\{\|w_t\|_1, 1\}$. Note that $A_t = \|a_t\|_1$.

$$\begin{aligned}
\|x_t - x_{t+1}\|_1 &= \left\| \frac{a_t}{A_t} - \frac{a_{t+1}}{A_{t+1}} \right\|_1 \\
&= \left\| \frac{(a_t - a_{t+1})A_{t+1} + a_{t+1}(A_{t+1} - A_t)}{A_t A_{t+1}} \right\|_1 \\
&\leq \frac{1}{A_t} \|a_t - a_{t+1}\|_1 + \frac{1}{A_t}(A_{t+1} - A_t) \leq 2 \|a_t - a_{t+1}\|_1.
\end{aligned}$$

$$\begin{aligned}
\|a_t - a_{t+1}\|_1 &= \left\| w_t + d^{-1} \max\{0, 1 - \|w_t\|_1\} - w_{t+1} - d^{-1} \max\{0, 1 - \|w_{t+1}\|_1\} \right\|_1 \\
&\leq \|w_t - w_{t+1}\|_1 + |\max\{0, 1 - \|w_t\|_1\} - \max\{0, 1 - \|w_{t+1}\|_1\}| \\
&\leq \|w_t - w_{t+1}\|_1 + |\|w_t\|_1 - \|w_{t+1}\|_1| \leq 2 \|w_t - w_{t+1}\|_1.
\end{aligned}$$

Therefore, $\|x_t - x_{t+1}\|_1 \leq 4 \|w_t - w_{t+1}\|_1$.

**Step 2** The second step is to add up the regret bound for each coordinates. Consider the $i$-th coordinate. Note that $|z_{t,i}| \leq 2G$. Using Theorem 6, for all $u_{1d} \in \mathbb{R}_+$,

$$\begin{aligned}
&\sum_{t=1}^{T} z_{t,i}(w_{t,i} - u_{1d}) + \tilde{\lambda} \sum_{t=1}^{T-1} |w_{t,i} - w_{t+1,i}| \\
&\leq \sqrt{(4\tilde{\lambda}\tilde{G} + 2\tilde{G}^2)T} \left[ \pi_i + |u_{1d} - \pi_i| \left( \sqrt{4\log\left(1 + \frac{|u_{1d} - \pi_i|}{\pi_i}\right)} + 2 \right) \right] \\
&= \sqrt{(32\lambda G + 8G^2)T} \left[ \pi_i + |u_{1d} - \pi_i| \left( \sqrt{4\log\left(1 + \frac{|u_{1d} - \pi_i|}{\pi_i}\right)} + 2 \right) \right].
\end{aligned}$$

Then, by summing up all the coordinates, for all $u \in \Delta(d)$,

$$\sum_{t=1}^{T} \langle g_t, x_t - u \rangle + \lambda \sum_{t=1}^{T-1} \|x_t - x_{t+1}\|_1$$

$$\leq \sum_{t=1}^{T} \langle z_t, w_t - u \rangle + 4\lambda \sum_{t=1}^{T-1} \|w_t - w_{t+1}\|_1$$

$$= \sum_{i=1}^{d} \left[ \sum_{t=1}^{T} z_{t,i}(w_{t,i} - u_i) + \tilde{\lambda} \sum_{t=1}^{T-1} |w_{t,i} - w_{t+1,i}| \right]$$

$$\leq \sqrt{(32\lambda G + 8G^2)T} \left[ 1 + 2\|u - \pi\|_1 + 2 \sum_{i=1}^{d} |u_i - \pi_i| \sqrt{\log\left(1 + \frac{|u_i - \pi_i|}{\pi_i}\right)} \right]$$

$$\leq \sqrt{(32\lambda G + 8G^2)T} \left[ 1 + 2\|u - \pi\|_1 + 2\sqrt{\|u - \pi\|_1} \sqrt{\sum_{i=1}^{d} |u_i - \pi_i| \log\left(1 + \frac{|u_i - \pi_i|}{\pi_i}\right)} \right].$$

(Cauchy-Schwarz)

Observe that since $u$ and $\pi$ both belong to $\Delta(d)$, $\|u - \pi\|_1 \leq 2$. If we define a function $f$ as

$$f := |x - 1| \log(1 + |x - 1|),$$

then using the standard definition of *f-divergence*

$$D_f(u||\pi) := \sum_{i=1}^{d} \pi_i f\left(\frac{u_i}{\pi_i}\right),$$

we have

$$\sum_{t=1}^{T} \langle g_t, x_t - u \rangle + \lambda \sum_{t=1}^{T-1} \|x_t - x_{t+1}\|_1 = \left[ \sqrt{\text{TV}(u||\pi) \cdot D_f(u||\pi)} + 1 \right] \cdot O\left( \sqrt{(\lambda G + G^2)T} \right).$$

**Step 3**  The last step is to upper bound $D_f(u||\pi)$ by $\text{KL}(u||\pi)$. To this end, notice that $\text{KL}(u||\pi) = D_g(u||\pi)$, where

$$g(x) := 1 - x + x \log x.$$

By Lemma C.1, $f(x) \leq 2g(x)$ for all $x \geq 0$, therefore $D_f(u||\pi) \leq 2D_g(u||\pi) = 2\text{KL}(u||\pi)$.   □

### C.3   Discussion on Algorithm 7

Here are some discussions to conclude our LEA analysis. First, the surrogate loss $z_t$ defined in Line 6 follows exactly the definition in [CO18, Algorithm 3]. We adopt this choice just to show the power of this general reduction technique. However, one could use other choices of $z_t$ and obtain the same guarantee, although the empirical performance could be different. For example, one can use

$$z_{t,i} = \begin{cases} g_{t,i} - \max_i g_{t,i}, & \text{if } \|w_t\|_1 < 1, \\ g_{t,i}, & \text{if } \|w_t\|_1 = 1, \\ g_{t,i} - \min_i g_{t,i}, & \text{if } \|w_t\|_1 > 1, \end{cases}$$

and clearly, the exact same proof still holds. Another possible choice is

$$z_{t,i} = \begin{cases} g_{t,i} - \sum_i g_{t,i}, & \text{if } \|w_t\|_1 < 1, \\ g_{t,i}, & \text{if } \|w_t\|_1 = 1, \\ g_{t,i} - \langle g_t, x_t \rangle, & \text{if } \|w_t\|_1 > 1. \end{cases}$$

This is more analogous to the surrogate losses in existing specialized approaches [LS15, OP16].

Also, to justify the improvement of $\sqrt{\text{TV} \cdot \text{KL}}$ over $\sqrt{\text{KL}}$, here is an example. For all $d \geq 3$, define $p, q \in \Delta(d)$ from

$$p_1 = \frac{1}{\sqrt{\log d}}, \quad q_1 = \frac{1}{d\sqrt{\log d}},$$

and

$$p_i = \frac{1 - p_1}{d - 1}, \quad q_i = \frac{1 - q_1}{d - 1}, \quad \forall i \in [2 : d].$$

Then,

$$\mathrm{TV}(p\|q) = \frac{1}{2}\left[|p_1 - q_1| + (d - 1)\left|\frac{1 - p_1}{d - 1} - \frac{1 - q_1}{d - 1}\right|\right] = |p_1 - q_1| = \frac{d - 1}{d\sqrt{\log d}},$$

$$\begin{aligned}
\mathrm{KL}(p\|q) &= p_1 \log \frac{p_1}{q_1} + (d - 1) \cdot \frac{1 - p_1}{d - 1} \log \frac{1 - p_1}{1 - q_1}\\
&= \sqrt{\log d} + \left(1 - \frac{1}{\sqrt{\log d}}\right)\log\left(1 - \frac{d - 1}{d\sqrt{\log d} - 1}\right)\\
&\geq \sqrt{\log d} + \log\left(1 - \frac{d}{d\sqrt{\log d} - 1}\right) = \sqrt{\log d} - o(1).
\end{aligned}$$

Since we also have

$$\mathrm{KL}(p\|q) = \sqrt{\log d} + (1 - p_1)\log\frac{1 - p_1}{1 - q_1} \leq \sqrt{d},$$

we can combine the above and obtain $\mathrm{TV}(p\|q) \cdot \mathrm{KL}(p\|q) \leq 1$ and $\mathrm{KL}(p\|q) \geq \sqrt{\log d} - o(1)$. If our comparator $u$ and prior $\pi$ take the value of $p$ and $q$ respectively, then even without switching costs, Theorem 2 saves a $(\log d)^{1/4}$ factor from the existing comparator adaptive bounds.

## D   Application to portfolio selection

To complement our theoretical results, we present applications to a portfolio selection problem with transaction costs.[7] Online portfolio selection has been studied by multiple communities, resulting in a large amount of literature (see [LH14, Doc16] for general expositions). Here we focus on an *unconstrained* setting, allowing both short selling (i.e., holding negative amount of assets) and margin trading (i.e., borrowing money to buy assets). This is related, but different from classical settings in the literature, as discussed in Appendix D.2.

### D.1   Problem setting

We consider a market with $d$ assets and discrete trading period $t \in \mathbb{N}_+$. In the $t$-th round, an algorithm chooses a portfolio vector $x_t = [x_{t,1}, \ldots, x_{t,d}] \in \mathbb{R}^d$, where $x_{t,i}$ is the *number of shares* of the $i$-th asset that the algorithm suggests to hold. Compared to the previous round, we need to buy $x_{t,i} - x_{t-1,i}$ shares[8] (or sell, if negative), which requires paying a $\lambda |x_{t,i} - x_{t-1,i}|$ transaction cost. Then, the market reveals a number $g_{t,i} \in [-G, G]$, which represents the price change per share (of the $i$-th asset) in this round. This effectively increases the value of our portfolio by $\langle g_t, x_t\rangle$.

The considered performance metric is the increased amount of *wealth* on any time horizon $[1 : T] \subset \mathbb{N}_+$, and such wealth includes the total value of our portfolio *plus cash*. Our goal is to show that the performance of our algorithm is never much worse than that of any unconstrained *Buy-and-Hold* (BAH) strategy, which picks a portfolio vector $u \in \mathbb{R}^d$ at the beginning and holds that amount throughout the considered time horizon. That is, we aim to upper bound $\sum_{t=1}^T \langle -g_t, x_t - u\rangle + \lambda \sum_{t=1}^{T-1} \|x_t - x_{t+1}\|_1$ for all $u \in \mathbb{R}^d$ and $T \in \mathbb{N}_+$. This is exactly the setting of our high dimensional OLO problem (Appendix B.2) with flipped gradients, therefore if we use our high dimensional OLO algorithm (Algorithm 6), then the same theoretical result (Theorem 7) carries over.

### D.2   Comparison to the rebalancing setting

The online portfolio selection problem has been studied both empirically and theoretically. Most theoretical works with adversarial guarantees consider the *rebalancing* setting, pioneered by Cover [Cov91] and followed by a series of works [CO96, HSSW98, KV02, OLL17, LWZ18, MR22, ZAK22]. Differences to our setting are discussed as follows.

---

[7]Code is available at https://github.com/zhiyuzz/NeurIPS2022-Adaptive-Switching.
[8]W.l.o.g., assume $x_0 = x_1$.

1. First, the rebalancing setting forbids short selling (i.e., $x_{t,i} < 0$) and margin trading (i.e., borrowing cash to buy an asset), therefore the decision is modeled as a distribution $p_t \in \Delta(d)$ – the algorithm redistributes its wealth according to this distribution in each round. In contrast, our setting allows both[9], so we call it "unconstrained". Similar to the loss-regret trade-off in OLO, allowing margin trading introduces a *risk-return trade-off* in some sense: based on its own risk tolerance, one can trade off the best-case return with the worst-case loss on a Pareto-optimal frontier.

2. Related to the above, existing works consider *Constant Rebalanced Portfolios* (CRP, i.e., $p_t = p^* \in \Delta(d)$) as the benchmark class, and the goal is to lower bound the *ratio* of the growth rate of the considered algorithm to the growth rate of the benchmark. Here we consider unconstrained *Buy-and-Hold* (BAH) strategies as benchmarks, and we aim at an additive bound on the wealth. There have been discussions on the correct choice of benchmarks, but as suggested by a series of works [Cov91, HSSW98, BK99], a major weakness of CRPs is the incorporation of transaction costs: such benchmark strategies lose money due to constant rebalancing in every round, which makes the performance guarantee vacuous in certain cases. In contrast, BAH benchmarks do not suffer from this issue.

3. Finally, transaction costs can take many forms. Here we consider the special case that charges a fixed price *per share*. This is different from the *proportional transaction cost* in some prior works [BK99, Gof14], which is proportional to the *total value* of the transaction.

We also note that our Algorithm 7 for LEA with switching cost is essentially a comparator adaptive improvement of the *Exponentiated Gradient* (EG) algorithm adopted in [HSSW98]. Therefore, it can be applied to the rebalancing setting, following the same argument there.

## D.3 Synthetic market

In this subsection, we present numerical results on synthetic markets.

Two algorithms are tested, our high dimensional OLO algorithm (Algorithm 6, i.e., "ours"), and the baseline from [ZCP22a] (its 1D version is surveyed as Algorithm 2 in Appendix A.1, and we extend it to high dimensions using the same coordinate-wise construction). Both algorithms require a confidence parameter $C$ – they are both set to 1 following the practice of comparator adaptive algorithms [OP16, CLO22, ZCP22a]. A detailed justification is provided later.

As for the synthetic market, we let $G = 1$, $\lambda = 0.1$, and the market contains five assets with different return characteristics. Each $g_{t,i}$ is the summation of a i.i.d. noise, a periodic fluctuation and a constant trend. Specifically, we consider three different market return models. The first is

$$g_{t,1} = 0.4 \cdot \mathrm{Uniform}[-1, 1] + 0.4 \sin[(t/500) \cdot \pi] + 0.2,$$
$$g_{t,2} = 0.5 \cdot \mathrm{Uniform}[-1, 1] + 0.3 \sin[(t/500 + 1/2) \cdot \pi] + 0.2,$$
$$g_{t,3} = 0.6 \cdot \mathrm{Uniform}[-1, 1] + 0.2 \sin[(t/500 + 1) \cdot \pi] + 0.2,$$
$$g_{t,4} = 0.7 \cdot \mathrm{Uniform}[-1, 1] + 0.1 \sin[(t/500 + 3/2) \cdot \pi] + 0.2,$$
$$g_{t,5} = 0.8 \cdot \mathrm{Uniform}[-1, 1] + 0.2.$$

The second model is

$$g_{t,1} = 0.2 \cdot \mathrm{Uniform}[-1, 1] + 0.4 \sin[(t/500) \cdot \pi] + 0.4,$$
$$g_{t,2} = 0.3 \cdot \mathrm{Uniform}[-1, 1] + 0.3 \sin[(t/500 + 1/2) \cdot \pi] + 0.4,$$
$$g_{t,3} = 0.4 \cdot \mathrm{Uniform}[-1, 1] + 0.2 \sin[(t/500 + 1) \cdot \pi] + 0.4,$$
$$g_{t,4} = 0.5 \cdot \mathrm{Uniform}[-1, 1] + 0.1 \sin[(t/500 + 3/2) \cdot \pi] + 0.4,$$
$$g_{t,5} = 0.55 \cdot \mathrm{Uniform}[-1, 1] + 0.45.$$

The third model is the same as the second one, except we replace $g_{t,5}$ by

$$g_{t,5} = 0.5 \cdot \mathrm{Uniform}[-1, 1] + 0.5.$$

For each market return model, we test both algorithms in 50 random trials, and the increased wealth $\sum_{\tau=1}^{t} \langle g_\tau, x_\tau \rangle - \lambda \sum_{\tau=1}^{t-1} \|x_\tau - x_{\tau+1}\|_1$ (mean ± std) is plotted in Figure 2, higher is better. In all three setting, our algorithm beats the baseline by a considerable margin, due to being a lot less conservative.

---

[9]Although we only consider the ideal case with zero interest rate on loans.

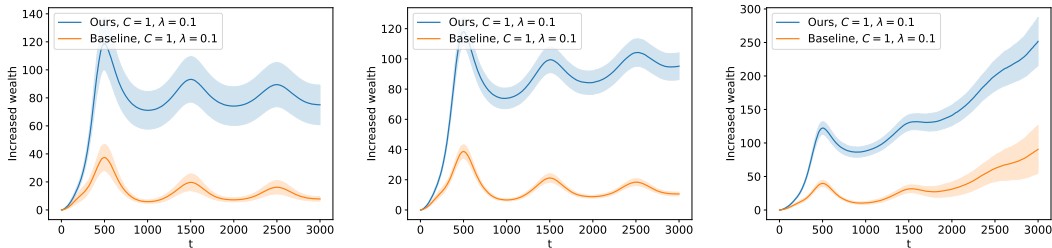

Figure 2: Synthetic market experiment with different market models. From left to right: the first, the second and the third market model.

**Discussion on** $C$    We remark that setting $C = 1$ for both algorithms may create confusion. Let us give it a detailed justification.

As surveyed in the Introduction, comparator adaptive algorithms are called "parameter-free" due to historical reasons. As the name suggests, one may question the existence of *any* hyperparameter in such "parameter-free" algorithms. The classical rationale is the following: comparator adaptive regret bounds depend on the hyperparameter $C$ logarithmically, whereas minimax regret bounds depend on the learning rate (and its inverse) linearly. In this regard, comparator adaptive algorithms are provably less sensitive to the correct setup, therefore as a rule of thumb, most practices [OP16, CLO22, ZCP22a] simply use $C = 1$ without requiring any domain knowledge. Such a default setup removes hyperparameter tuning, which is the most attractive feature of these algorithms in practice. Figure 2 shows the advantage of our algorithm when both algorithms are in this default, parameter-free implementation.

Nonetheless, for specific tasks like portfolio selection, tuning $C$ can affect the actual performance one cares about (although violating the main purpose of comparator adaptivity). Intuitively, fixing the market, an aggressive trader with a worse strategy could make more profit than a conservative trader with a better strategy. Reflected in our experiment, since the market model does not depend on the invested amount, the baseline with a 5 times larger $C$ simply obtains a 5 times larger increased wealth and beats our algorithm (at certain $t$), cf. Figure 3 (Left). Of course, one can also tune our algorithm with a correspondingly scaled $C$ and beat the baseline again, cf. Figure 3 (Right), just like when both algorithms are in their parameter-free implementation.

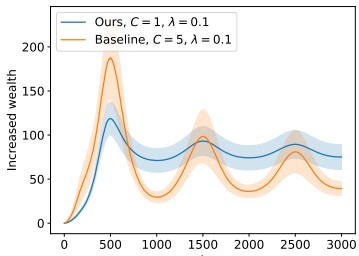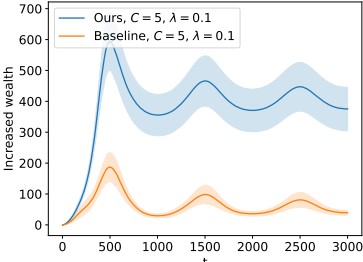

Figure 3: Synthetic market experiment with tuned $C$; not a parameter-free implementation. Left: only tuning the baseline. Right: tuning both our algorithm and the baseline.

Therefore, if tuning $C$ is allowed, then comparing our algorithm to the baseline amounts to comparing two *algorithm classes* both parameterized by $C$. A skeptical reader may wonder if the superior performance of our algorithm in the parameter-free setting is due to the confidence encoding rather than a better algorithm design. That is, is it possible that a baseline with a larger $C$ can consistently outperform our algorithm with $C = 1$? We provide evidence against this hypothesis, by increasing $\lambda$ while keeping $C = 1$ for our algorithm and $C = 5$ for the baseline; results are plotted in Figure 4. It shows that even when the baseline is given an advantage ($C = 5$), our algorithm is still better at handling transaction costs due to an improved design. This is aligned with the superiority of our theoretical results.

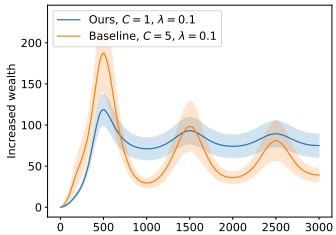 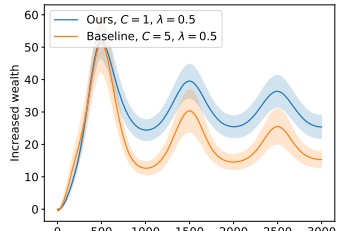 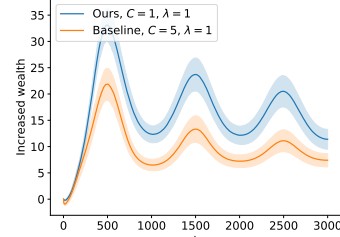

Figure 4: Synthetic market experiment with increasing $\lambda$. Left: $\lambda = 0.1$. Middle: $\lambda = 0.5$. Right: $\lambda = 1$. The baseline is given an advantage ($C = 5$), while our algorithm is in its default parameter-free implementation ($C = 1$). It shows our algorithm indeed handles transaction costs better.

## D.4 Historical stock data

Finally, we present some preliminary numerical results on historical US stock data[10]. Eight stocks (Table 1) are considered on a time period of 5 years (1/1/2013 to 1/1/2018). Our algorithm trades once per day after the market closes, based on the closing price. We take the difference between the closing price on the $(t + 1)$-th day and the closing price on the $t$-th day, and define it as the market vector $g_t$. The largest single day price change for any stock is less than \$15, therefore $G$ is set in a posterior manner to 15. We consider a hypothetical broker that charges \$0.1 per share, therefore define $\lambda = 0.1$.

Table 1: List of considered stocks

| Company | Symbol |
|---|---|
| Apple Inc. | AAPL |
| Berkshire Hathaway Inc. Class B | BRK.B |
| Meta Platforms Inc. | FB |
| Johnson & Johnson | JNJ |
| JPMorgan Chase & Co. | JPM |
| Microsoft Corporation | MSFT |
| Pfizer Inc. | PFE |
| Exxon Mobil Corporation | XOM |

Same as the synthetic market experiment, we test our algorithm against the baseline from [ZCP22a]. Our algorithm is in its default parameter-free implementation ($C = 1$). However, setting $C = 1$ also for the baseline is too conservative, which means the baseline hardly makes any investment, making the comparison less interesting. Therefore we set $C = 10$ for the baseline, thus giving it an advantage at the beginning. In this way, the increased wealth of the two algorithms is roughly comparable.

We plot the results in Figure 5. Specifically, Figure 5 (Left) shows the increased wealth (in USD) over the considered time period. Figure 5 (Right) shows the cumulative amount of investment (in USD), which is the total net amount of cash the investor uses to buy stocks (i.e., increases when buying, and decreases when selling), plus the transaction costs paid to the broker. Before analyzing this result, we note that such a "cumulative investment" only makes sense in our setting, due to a fundamentally different mechanism compared to the rebalancing approach [Cov91]: in the latter, the investor is *self-financed*, i.e., it is given a certain budget at the beginning and never adds more money from external sources after that. In contrast, the investor in our setting can add more money at any time it wishes.

From the plot we can see that the baseline is more aggressive at the beginning, due to a much larger $C$. Therefore, it slightly makes more profit during 2013-2014. When the market oscillates and declines in 2015 and 2016, the two algorithms perform roughly the same, while the baseline has a lower risk due to holding a smaller portfolio at the time. However, the major difference starts after mid-2016, when the market grows rapidly. Our algorithm is able to identify this trend and quickly increase

---

[10]US stock price data is publicly available. We retrieved the data from Yahoo Finance website. `https://finance.yahoo.com/`

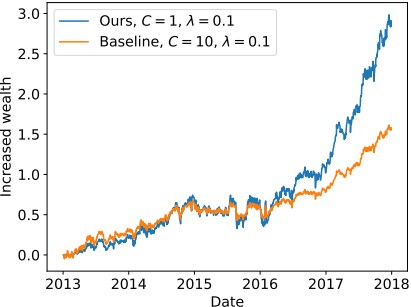 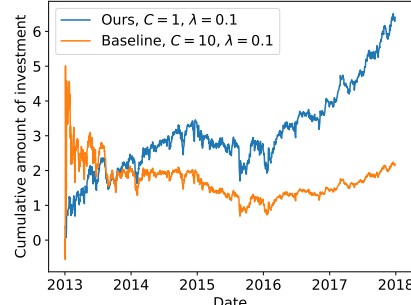

Figure 5: Experiment on historical US stock data. Left: the increased wealth of the two algorithms. Right: total amount of investment since the start of the experiment (1/1/2013), including the transaction costs paid to the broker.

the amount of investment. This brings a lot more profit than the baseline, which hardly recovers its confidence from the declining market in the previous year. Such an advantage of our algorithm is partly due to the better control of switching costs, and partly due to a better risk-return trade-off discussed in Appendix A.4.

Our experiment also shows a limitation of our unconstrained investment setting. Throughout this five year period, our algorithm invests a total amount of $\sim$\$6.5 (including the transaction costs), and makes a total profit of $\sim$\$3. However, in practice, one typically invests a lot more than this (let's say, \$10,000), and expect a similar rate of return. Our setting does not model such a budget explicitly; instead, it relies on the comparator adaptivity of the trading algorithms to increase the invested amount. Such a process can be slow, especially since we only consider trading once per day. Therefore, to use our algorithm in real trading situations, one has to tune the confidence parameter $C$ to implicitly take his budget and tolerable risk into account. For example, using our algorithm with $C = 1000$ would result in investing \$6,500 throughout the five year period, and make a total profit of \$3,000. The connection of this approach to rebalancing could be an interesting direction for future works.

# E Conclusion, limitation and future work

The present work investigates the design of comparator adaptive algorithms in the presence of switching costs. By carefully trading off these two opposite considerations, we propose a simple algorithm for OLO with switching costs, improving the suboptimal bound from our prior work [ZCP22a] to the optimal rate. Notably, the key idea of this algorithm is not guessed, but derived from a continuous-time analysis. Extensions lead to new results for comparator adaptive LEA.

**Limitation and future work** Our result requires a time-invariant $\lambda$, which could be generalized in future works. Different from [ZCP22a], we did not discuss applications to control theory, which is interesting on its own. Also, one may combine our portfolio selection approach with adversarial rebalancing and stochastic modeling, in order to further improve its practical performance.

More generally, through this paper we aim to demonstrate a key strength of the continuous-time PDE analysis – it makes the generalization of algorithmic structures much easier. Such an observation could open up exciting possibilities:

- Does this approach apply to other variants of the online learning problem?
- Can we use it to generalize other forms of adaptivity?
- Continuous-time potentials have been extensively studied under the framework of *potential theory* [Doo84]. Can we borrow techniques from there to further improve the workflow of algorithm design?