# OpenReview forum: "Optimal Comparator Adaptive Online Learning with Switching Cost"
_NeurIPS.cc/2022/Conference — NeurIPS 2022 Accept_

### Official Review · Reviewer_NFWC · 2022-07-06

**Rating:** 7
**Confidence:** 1
**Soundness:** 3 good
**Presentation:** 1 poor
**Contribution:** 3 good

**Summary:**

This theoretical article is about "parameter-free" regret minimization for (linear) online learning with switching cost

For non-specialists like me, this appealing notion of "parameter-free" needs to be clarified somehow. When analysing an algorithm like stochastic gradient descent, its convergence time upper bound is often given according to some hyper-parameters like the learning rate. The best convergence time is then obtained by setting the learning rate that minimizes this convergence time. If this learning rate is unknown, tuning this parameter has a cost which impacts the convergence time. It has been shown that his cost depends on the square root of the logarithm of the distance between the initial point of the optimization and the (unknown) target.
It seems however that some "parameters" of the problem, like the Lipschitz constant G, that are required by the algorithms are not taken into account in this definition. In other words "parameter-free" means that we integrate the cost of tuning -- some -- parameters in the analysis.

The ability for an online algorithm to handle switching costs is appealing for practical applications where mixing policies is costly or not desirable. By essence, a parameter-free algorithm has to be more adaptable hence more reactive than a classical algorithm, but the switching cost favors algorithms with stable behaviours. This leads to a balance

The key contribution of this article is to provide a new dual approach that improves the parameter-free regret bound provided in [Zhang et al. 2022a] for this switching cost setting.

The Online Linear Optimization algorithm is then transposed to solve the Online Learning with Expert's Advices problem (with switching cost). And it is evaluated through simulations on a portfolio selection problem with transaction costs. It compares favorably against the [Zhang et al. 2022a] baseline.

The proofs are detailed along 20 pages of supplementary material. Some code is provided as well.

**Questions:**

The algorithm being not really parameter-free, why not change the title to "Optimal almost-parameter-free Online Learning with Switching Cost" ?
You set the G and lambda parameters respectively to G=1 and lambda=0.1 in your synthetic market experiments. Did you inform your algorithm of these parameters beforehand ? Did you try running your algorithm with the wrong parameters ?
Do the coin betting strategy proposed in [Zhang et al. 2022a] require such parameters ? Is the comparison fair on that ground ?


**Limitations:**

The limitations are explicitly given in the conclusion.
The authors underline that the Lipschitz constant G and the time-invariant switching cost lambda are required by their algorithms. Hence admitting that their algorithms are not really "parameter-free".

Adapting to changing costs and dropping the G parameter is left as future work.


**Strengths And Weaknesses:**

Weaknesses / General remarks:
- The paper is far from being self-contained. It has been written for researchers who are already well read on "parameter-free" online learning. As a newcomer I had to follow a full tutorial on the subject to understand the introduction. I think the author should make an effort to write a more readable introduction. Maybe by providing a few concrete historical examples of parameter-free algorithms.
- The "parameter-free" appellation used by this community is "historic" but still a bit of an overstatement for an algorithm, like Algorithm 1, which requires to know or guess the Lipschitz constant G and the switching cost lambda parameter in order to work.

Strengths:
- The integration of switching costs in the Online Learning with Expert's Advices (LEA) problem is an important problem both from the practical point of view and from the theoretical point of view.
- An efficient online learning algorithm able to cope with this problem with as few parameters as possible is a clear contribution (even if G and lambda are still required).
- Once into it, the paper is correctly written, and the maths seem solid (but I did not check the proofs).
- The author provided a code to reproduce their experiments in supplementary material.

---

> ### Author Response · Authors · 2022-08-01
> **Response to Reviewer NFWC**
>
> Thank you for your careful review, we appreciate your general support for our paper.
>
> 1. On the writing. Thank you for your suggestions. We agree with you that our introduction is a bit technical and requires some prior knowledge on parameter-free online learning. We will try to reduce the amount of technical arguments and make it more approachable for a broader group of readers. Also, we will add a section to the appendix surveying the background of this topic.
>
> 2. On the terminology. Yes, we totally agree that the name "parameter-free online learning'' is due to certain historical reasons, and it is kind of an overclaim. Recent works often use "parameter-free online learning'' and "comparator-adaptive online learning'' interchangeably. Based on your comment, we think the latter would be more accurate, especially for future research building on our work. We will revise this in the camera-ready version.
>
> 3. Questions on the experiment. Both our algorithm and the baseline [ZCP22a] require $G$ and $\lambda$ as inputs. In our experiment, we feed the correct $G$ and $\lambda$ to them, thus making the comparison fair. The result qualitatively verifies our theory. However, we did not try incorrect $G$ and $\lambda$ on these algorithms. This is a bit orthogonal to the focus of this paper, but indeed, we think the robustness of these algorithms is an interesting problem for future research (both theoretically and empirically).

---

### Official Review · Reviewer_PZQr · 2022-07-09

**Rating:** 5
**Confidence:** 4
**Soundness:** 3 good
**Presentation:** 3 good
**Contribution:** 2 fair

**Summary:**

This paper studies the problem of parameter-free online learning in the presence of switching cost. Based on a novel dual space scaling strategy, the authors obtain the optimal regret bound for unconstrained OLO with switching cost. The results are also extended to the expert setting, and numerical results on portfolio selection problem are reported.

**Questions:**

In this paper, the switching costs are in the form of $L_{1}$ norm and the authors claim that the proposed strategy can be extended to other norms. Since $L_{2}$ norm or $L_{2}$ norm squared are also widely used for switching costs, I wonder how to extend the strategy to $L_{2}$ norm, especially for higher-dimensional domains?

(I mean, for higher dimensions, the authors run Algorithm 1 on each coordinate separately to solve it. This is valid for $L_{1}$ norm, since the $L_{1}$ switching cost is the sum of the cost of each coordinate. However, would it be problematic for $L_{2}$ norm or $L_{2}$ norm squared?)

**Limitations:**

No problems here.

**Strengths And Weaknesses:**

Strengths:

1. The proposed algorithm is parameter-free and improves the suboptimal result in [ZCP22a] to the optimal rate. The results can also extend to the expert setting.

2. This paper is clear-written and technically sound. The theoretical analyses are supported by rigorous proofs which are sound to me.

Weakness:

My main concerns are contribution and novelty.
1. As for contribution, since there already exist methods such as [ZJLY21], that achieve optimal $\mathcal{O}(\sqrt{T})$ (dynamic) regret for OCO problem. Although the proposed method in this paper does not require the diameter of the domain, this improvement seems not significant enough.

2. As for novelty, the proposed method and the proofs are not very different from [ZCP22b], which have already been mentioned in the paper. So, it seems to me that the method is not novel enough.

---

> ### Author Response · Authors · 2022-07-31
> **Response to Reviewer PZQr**
>
> Thank you for your feedback. However, with due respect, we believe the main contribution of our paper and its novelty have been overlooked in your review.
>
> **1. "Comparison to other online learning papers with switching costs."**
>
> Online learning with switching costs has been studied by many prior works, as we reviewed in Line 101-116. However, in this paper we study **adaptive (parameter-free) algorithms** in this setting. **This is significant and technically nontrivial for many reasons.**
>
> - Non-adaptive algorithms require a bounded domain, and typically they (including [ZJLY21]) use the diameter of the domain to tune the learning rates. However, many practical problems are naturally defined on unconstrained domains where *non-adaptive algorithms cannot be applied*. We not only relax the assumption of knowing the diameter, but also deal with the more practical and challenging setting where the domain is not bounded at all.
>
> - Parameter-free algorithms can take a prior as initialization, and the regret bound naturally adapts to it. This is useful when a good prior can be obtained from domain knowledge or transfer learning. Furthermore, in the LEA setting (Line 85-93), parameter-free algorithms are automatically equipped with quantile regret bounds.
>
> - Parameter-free algorithms with switching costs have important applications in linear control, as reviewed in Line 123-131 and highlighted by Reviewer KR3f's comment.
>
> - Algorithmically, parameter-free algorithms are based on very different principles compared to non-adaptive ones. For example, they typically do not have learning rates. The design of these algorithms is less understood, and especially, the problem becomes even more challenging when switching costs are introduced. Our paper has to deal with the trade-off between parameter-freeness and switching costs (Line 40-51), and it is not clear a priori what is the optimal rate and how to achieve it.
>
> **2. "Comparison to [ZCP22b]."**
>
> Our algorithm belongs to the classical potential method in online learning. The main contribution of [ZCP22b] is a continuous-time framework for designing parameter-free potentials without switching costs. **In this paper, we extend their argument in a nontrivial way:**
>
> For our setting, the only known algorithm [ZCP22a] is not a potential method - it is unclear whether any potential method can achieve our goal. A suitable potential function should incorporate the switching cost weight $\lambda$ in a smart way. The question is, how to do this? Simply scaling the learning rates does not work, since here we don't have learning rates at all.
>
> It turns out that bringing the game to the continuous-time limit (Appendix A.1) indeed yields a good potential, and more importantly *gives us an interpretable insight*: the key is to use $\lambda$ to scale the sufficient statistic, and the scaling order also naturally emerges. In other words, besides the argument from [ZCP22b], we further show that
>
> **- the continuous-time framework could design potential functions for challenging settings where no potentials have been known to work;**
>
> **- this framework not only is quantitative, but also generates interpretable insights.**
>
> We believe these features are nontrivial contributions to the continuous-time framework for adaptive online learning [ZCP22b].
>
> As for the techniques, we only use the framework of [ZCP22b] to derive the continuous-time potential. Verifying this potential in discrete time is a different task - we need to use a more conservative $\alpha$ than what is suggested in continuous time. Moreover, the two key lemmas (Lemma 2.2 and 2.3) for the analysis are technically nontrivial and not contained in [ZCP22b].
>
> **3. "$L_2$ switching cost."**
>
> The prior work [ZCP22a] considers $L_2$ switching cost. Its Alg.2 is a reduction from the general dimensional problem to 1D, based on the standard polar-decomposition trick. The idea is to parameterize the high-dimensional space using a direction and a length - the direction can be learned using gradient descent on the unit $L_2$ norm ball, and the length can be learned using the 1D parameter-free algorithm on $\\mathbb{R}_+$.
>
> Our algorithm directly improves the 1D algorithm of [ZCP22a], therefore can also use the above technique to handle higher-dimensional $L_2$ switching costs. The only required modification is an extra regularization term. That is, we have to bound
> $$
> \sum_{t=1}^Tg_t(x_t-u)+\lambda\\sum_{t=1}^{T-1}|x_t-x_{t+1}|+\sum_{t=1}^T\frac{\lambda}{\sqrt{t}}|x_t|.
> $$
> To do this, we can define a surrogate loss $l_t(x)=g_tx+\lambda t^{-1/2}|x|$, and then run our algorithm on the linearized version of $l_t$. It is not hard to verify that the regularization term can also be controlled in this way. We will add this extension to the camera-ready version.
>
> Hope our response has addressed your concerns. If so, we would appreciate it if you update your rating. We would also be glad to answer further questions.

---

> > ### Comment · Reviewer_PZQr · 2022-08-04
> > **Thanks for the response!**
> >
> > Thank you for the detailed response.
> >
> > Although I still worry that the techniques used in this paper is somewhat incremental to [ZCP22b], the response indeed address some of my concerns.
> > I am willing to revised my score from 4 to 5.

---

### Official Review · Reviewer_1xnT · 2022-07-11

**Rating:** 6
**Confidence:** 3
**Soundness:** 3 good
**Presentation:** 3 good
**Contribution:** 3 good

**Summary:**

The authors design an algorithm for online linear optimization (OLO) with switching cost, and prove the optimality of the proposed algorithm, as an improvement of an existing algorithm. The trade-off between adaptivity and switching cost, as well as an extension of the algorithm to the learning with expert advice (LEA) are discussed.

**Questions:**

1. I am a bit confused by the argument from line 156 to 163. Suppose the sum of switching cost is $1+|u|O(\sqrt{T\log(|u|T)})$, then this "budget" reduces to $O(1)$ only when $u=0$, which implies the comparator is not far away from the initial $x$. For far-away comparators, the budget grows at $O(\sqrt{T})$. I do not see how the conclusion is drawn.

2. In Algorithm 1, the authors set a hyperparameter $C$, and there is also another parameter $\alpha$ in the potential function. Will the performance be sensitive to these two parameters? Should one always choose $\alpha = 4\lambda G^{-1}+2$ in practice?

3. Is it easy and fast to compute the discrete derivative $\bar{\nabla}_SV(t,S)$ in Algorithm 1, when the dimension is not small?

4. What does $||\cdot||_*$ stand for? (e.g., in line 94)

**Limitations:**

The authors adequately discuss the limitation and potential improvement for the paper.

**Strengths And Weaknesses:**

Strengths:
1. While the problem of OLO with switching cost is not new, the authors improve the result on the regret bound, and show the optimality of their result in several forms.
2. The key idea that incorporating switching costs by scaling on the dual space seems new and interesting.
3. Overall the paper is well-organized and mostly easy to follow.

Weakness:
1. As authors point out, the result requires the knowledge of the Lipschitz constant G and assumes $\lambda$ is time-invariant, which might limit the application of the method in practice.
2. Please see my additional questions/concerns in the next part.

---

> ### Author Response · Authors · 2022-07-31
> **Response to Reviewer 1xnT**
>
> Thank you for your feedback and your support for our paper.
>
> **1. Question on Line 156-163.**
>
> The argument we aim to make is the following. Suppose we can show that the two terms in Eq. (2) *separately* satisfy the parameter-free bound. Then, it means that for all $u\in\mathbb{R}$,
> \begin{equation*}
> \sum_{t=1}^{T-1}|x_t-x_{t+1}|\leq 1+|u|\tilde O(\sqrt{T}).
> \end{equation*}
> The left hand side only depends on the operation of the algorithm, and *does not depend on the comparator $u$*. Therefore, in order for the above to hold for all $u\in\mathbb{R}$, we must have $\sum_{t=1}^{T-1}|x_t-x_{t+1}|\leq 1$, which is clearly very conservative.
>
> **2. "Sensitivity to $C$ and $\alpha$."**
>
> The dependence on $C$ is quite standard in parameter-free online learning. Setting it to an arbitrary constant (e.g., 1) is already good enough for theoretical purpose and many practical applications [CLO20].
>
> As discussed in Appendix A.1, the ideal $\alpha$ suggested by the continuous-time analysis is $1/2+\lambda$. We use a larger one just to control the discretization error (Lemma 2.3), but this is a sufficient condition rather than a necessary one. In practice we speculate that setting it slightly lower would still lead to good performance.
>
> **3. "Computational efficiency."**
>
> Yes, computing the discrete derivative is easy and fast. The potential function in its double integral form may seem ugly, but due to Line 764, we can rewrite it without integrals, using the *imaginary error function* $\mathrm{erfi}$. Evaluating the latter is easy in standard software packages like scipy.
>
> For general $d$-dimensional problems, we run a 1D algorithm on each dimension. This is the same idea as AdaGrad, and isn't hard to compute. Furthermore, we can use a simple polar-decomposition technique as in [ZCP22a], such that we only need one 1D algorithm instead of $d$ copies of it.
>
> **4. Notation**
>
> We apologize for the confusion, $||\cdot||_*$ means dual norm. We will add a definition to it.

---

### Official Review · Reviewer_KR3f · 2022-07-11

**Rating:** 6
**Confidence:** 4
**Soundness:** 3 good
**Presentation:** 3 good
**Contribution:** 2 fair

**Summary:**

This work studies a meaningful problem called online linear optimization with switching costs. Previous progress on this problem, [ZCP22a], obtains a suboptimal regret bound through a complicated coin betting algorithm. This work improves the previous result in several aspects with a simpler and more elegant algorithm based on some specially designed potential functions. This work also extends the above result to higher dimensions and bounded domains. Besides, in LEA with switching cost, with more structural information, this work proposes a novel reduction from bounded to unbounded domains, which improves the previous bound even without the existence of switching cost. Finally, empirical studies evaluate the effectiveness of the proposed method.

**Questions:**

1. Actually, [ZCP22a] already obtained an optimal dependence on $\lambda$, i.e., $O(\sqrt{\lambda})$, through a mini-batching technique (see Algorithm 3 in [ZCP22a] for more details), but the authors do not mention it in the draft. Can the authors explain about this?

2. The work [ZCP22b] seems to share much in common with this work, such as potential functions. While in my opinion, the authors did not sufficiently compare the difference between this work and that of [ZCP22b], which is inappropriate. Can the authors give a more detailed comparison of their work and [ZCP22b], in terms of both problems and techniques?

3. Although the main algorithm, i.e., Algorithm 1, is pretty simple and thus elegant, it is not intuitive enough, at least for me. For example, what is the intuition behind the potential function in (3)? The authors said that there is a corresponding analysis in Appendix A.1. Can the authors explain more about the intuition that guides the algorithm design?

4. I guess the extension to online non-stochastic control as in [ZCP22a] is straightforward by reducing the control problem to OCO with memory with a particular policy parametrization?

5. The problem of OLO with switching cost is meaningful due to its deep connection with some online decision-making problems, such as online non-stochastic control. As a suggestion, the authors can make this connection clearer in the main paper, at least in the introduction part.


**Ethics Review Area:**

["I don’t know"]

**Limitations:**

Not much.

**Strengths And Weaknesses:**

Strengths:

1. The problem of OLO with switching cost is meaningful due to its deep connection with some online decision-making problems, such as online non-stochastic control.

2. The main algorithm, i.e., Algorithm 1, is simple and elegant. And the corresponding theoretical result improves the suboptimal one in [ZCP22a] in multiple aspects.

3. The result in LEA with switching cost is novel, which also imports some novel ideas in the LEA-OLO reduction, even without the existence of switching cost. The illustration in Figure 1 is clear enough.

Weaknesses: please mainly refer to the “Questions” part below.

---

> ### Author Response · Authors · 2022-07-31
> **Response to Reviewer KR3f**
>
> First, thank you for your insightful comments!
>
> **1. "The baseline can achieve $O(\sqrt{\lambda})$ rate through a mini-batching technique."**
>
> This is completely correct, thank you for spotting it! We were focusing on the comparison to the vanilla form of the baseline (Alg.1 of [ZCP22a]) and overlooked the extension in Alg.3 of the latter. This is our fault, and we will correct it in the final version. Our two main improvements over the baseline still hold, i.e., Pareto-optimal loss-regret trade-off and the overall $O(\sqrt{T})$ rate (even if one insists on constant regret at the origin, by setting $C$ appropriately we improve the logarithmic factor of [ZCP22a] from $\log(T)$ to $\sqrt{\log(T)}$, which is optimal).
>
> **2. "Relation to [ZCP22b]."**
>
> Our paper and [ZCP22b] have quite different scopes, therefore their comparison is not emphasized in the Introduction. Specifically,
>
> **Setting.** [ZCP22b] considers the parameter-free OLO problem *without switching costs*, therefore it does not face the challenge of trade-offs described in Line 45-51. Instead, [ZCP22b] considers the same setting as [MO14] and [OP16], but with improved bounds. Line 94-100 in our paper surveys this line of work.
>
> **Contribution.** [ZCP22b] makes two main contributions: ($i$) improving the standard parameter-free OLO bound [OP16]; and ($ii$) proposing a continuous-time framework to design parameter-free potentials. Their first contribution is not applicable to us since our setting is different. **In this paper we further develop their second contribution in a nontrivial manner, as described below.**
>
> First, without switching costs, the potential method is a popular strategy for parameter-free online learning, and [ZCP22b] provides a good way to design these potential functions. However, when switching costs are considered, it is not so clear whether the potential method is still effective - the existing algorithm [ZCP22a] is not a potential method. Suppose a potential method works in this setting, then apparently it needs a good way to incorporate the switching cost weight $\lambda$. The question is, how to do this? Simply scaling the learning rates (as in the minimax analysis of switching costs) does not work, since here we *don't have learning rates at all*.
>
> It turns out that bringing the game to the continuous-time limit (Appendix A.1) indeed yields a good potential, and more importantly *gives us an interpretable insight*: the key is to use $\lambda$ to scale the sufficient statistic, and the scaling order also *automatically* emerges. In other words,
>
> **- the continuous-time framework could design potential functions for challenging settings where no potentials have been known to work;**
>
> **- this framework not only is quantitative, but also generates interpretable insights.**
>
> We find these features quite remarkable, and nontrivially extend the argument from [ZCP22b].
>
> **Technique.** As discussed in Appendix A.1, our derivation of the potential function has a similar intuitive flow as [ZCP22b]. However, there are major deviations when we convert the analysis back to discrete time, discussed in Line 602-607. We need to set $\alpha$ to a more conservative value. Moreover, the key Lemmas 2.2 and 2.3 are technically nontrivial, and not contained in the prior work.
>
> **3. "Intuition behind the continuous-time analysis and the potential (3)."**
>
> Our analysis has the following intuitive flow. We start from Eq (5) in Line 568 - this is a discrete-time recursion characterizing a class of good potentials. *Any* function $V$ that satisfies this recursion would yield a regret bound, therefore our goal is to solve this recursion and find the right potential function among the solution class. However, solving this discrete-time recursion in closed-form is a challenging task, therefore instead of pursuing exact solutions, we pursue approximate solutions in continuous time. This still leads to regret bounds, as long as we characterize the approximation error.
>
> We approach continuous time by scaling the unit time and the gradient space in Eq (5), and this gives us a PDE (Line 588). Quite surprisingly, this PDE is the same type as the one from [ZCP22b] but with a different coefficient. Put in another way, incorporating the switching costs has such a simple quantitative effect in the continuous-time limit, which is quite hard to observe in the original discrete-time setting.
>
> Finally, we use a *change-of-variable* to transform our PDE into the one from [ZCP22b]. This naturally yields the interpretable insight of *dual space scaling*.
>
> **4. "Extension to non-stochastic control.''**
>
> Yes, our algorithm can be used to construct strongly adaptive algorithms for OCO with memory. A couple of linear control algorithms can stem from there. We also appreciate your suggestion on writing. Adding a discussion on nonstochastic control can indeed motivate our problem better.

---

> > ### Comment · Reviewer_KR3f · 2022-08-07
> > **Thanks for the response.**
> >
> > I think the authors have properly addressed my questions. It would be very nice if the authors could incorporate those discussions in the revised version. So I will increase my score to acceptance.

---

### Official Review · Reviewer_cFUB · 2022-07-11

**Rating:** 6
**Confidence:** 1
**Soundness:** 3 good
**Presentation:** 3 good
**Contribution:** 3 good

**Summary:**

The authors study online learning in a parameter-free setting. These settings have the benefit of being applicable to settings where the domain is unbounded and where prior information is not available. They generalize some existing algorithms to include switching costs. Switching costs can model a scenario where the learner is incentivized to not change its prediction significantly. They give an algorithm for Online linear optimization that is more or less optimal in several useful metrics. They also experiment on both synthetic datasets as well as real-world datasets, surpassing the baseline significantly.


**Questions:**

The result for Algorithm 1 is when G is exactly known. Quantitatively, how much does the bound worsen if only an upper bound on G is known?


**Limitations:**

None.

**Strengths And Weaknesses:**

Originality & Significance -  The models and framework seem to be standard.  The work done in the paper seems to be original to the best of my limited knowledge. Related to the prior work, I find the current work to be incremental and is of moderate significance.

Quality & Clarity - The paper is well-written and the exposition is easy to follow.  The quality of writing is above-average and the arguments made in the non-technical parts of the paper were cogent. Due to my lack of domain knowledge and time constraints I did not verify all technical details of the paper.

---

> ### Author Response · Authors · 2022-07-31
> **Response to Reviewer cFUB**
>
> Thank you for your feedback, we appreciate your support for our paper.
>
> Regarding the novelty of our paper, we respectfully disagree with your comment that it is incremental compared to prior works. Essentially, parameter-freeness and switching costs are two opposite considerations we need to trade-off. Such trade-offs in online learning are quite subtle, and it is generally unclear where the achievable boundary is (Line 45-51). The prior work [ZCP22a] only achieved a suboptimal trade-off using a quite complicated strategy, which isn't satisfactory both theoretically and empirically. In this paper we show that a much simpler strategy, when designed carefully, can indeed improve the bound to the optimal rate. Moreover, it reveals an elegant insight: a general way of incorporating switching costs is to scale on the dual space. We believe this is a nontrivial contribution to the field.
>
> Regarding your question, "how much does the bound worsen if only an upper bound on $G$ is known?'' Our bound would scale linearly with the given Lipschitz constant $G$. However, we may do better by also considering *the adaptivity to the observed gradients* - this is an interesting open problem for future research. When there are no switching costs, there exist parameter-free algorithms (e.g., [MK20]) that ($i$) do not require a given $G$; and ($ii$) guarantee $\tilde O(|u|\sqrt{\sum_{t=1}^T|g_t|^2})$ bound instead of $\tilde O(|u|G\sqrt{T})$. The problem becomes trickier when we add switching costs, as hinted by some negative results [Gof14]. It is unclear where the achievable boundary is.

---

### Meta-Review · Area_Chair_vNvw · 2022-09-01

**Recommendation:** Accept
**Confidence:** Certain

**Metareview:**

This is a technical, but interesting paper on online linear optimization. The nice contribution is a control of the switching cost (moving from one action to another) which makes the problem highly non-trivial.

The contribution is to consider a "smaller" set of assumptions (hence a weaker asymptotic result) than in the existing literature, but this allows to get better parametric rates.

This might not be the most breathtaking paper, but the reviewers and myself find it sufficiently interesting to be accepted at NeurIPS.

Congratulations !

**Award:**

No

---

### Decision · Program_Chairs · 2022-09-14

Accept